# Cryo-EM structures of engineered active $bc_1$-$cbb_3$ type CIII$_2$CIV super-complexes and electronic communication between the complexes

Stefan Steimle[1], Trevor van Eeuwen [2], Yavuz Ozturk[1,3], Hee Jong Kim [2], Merav Braitbard[4], Nur Selamoglu[1], Benjamin A. Garcia[5], Dina Schneidman-Duhovny[4], Kenji Murakami [5✉] & Fevzi Daldal [1✉]

Respiratory electron transport complexes are organized as individual entities or combined as large supercomplexes (SC). Gram-negative bacteria deploy a mitochondrial-like cytochrome (cyt) $bc_1$ (Complex III, CIII$_2$), and may have specific $cbb_3$-type cyt $c$ oxidases (Complex IV, CIV) instead of the canonical $aa_3$-type CIV. Electron transfer between these complexes is mediated by soluble ($c_2$) and membrane-anchored ($c_y$) cyts. Here, we report the structure of an engineered $bc_1$-$cbb_3$ type SC (CIII$_2$CIV, 5.2 Å resolution) and three conformers of native CIII$_2$ (3.3 Å resolution). The SC is active in vivo and in vitro, contains all catalytic subunits and cofactors, and two extra transmembrane helices attributed to cyt $c_y$ and the assembly factor CcoH. The cyt $c_y$ is integral to SC, its cyt domain is mobile and it conveys electrons to CIV differently than cyt $c_2$. The successful production of a native-like functional SC and determination of its structure illustrate the characteristics of membrane-confined and membrane-external respiratory electron transport pathways in Gram-negative bacteria.

[1] Department of Biology, University of Pennsylvania, Philadelphia, PA, USA. [2] Biochemistry and Molecular Biophysics Graduate Group, Perelman School of Medicine, University of Pennsylvania, Philadelphia, PA, USA. [3] Institute of Biochemistry and Molecular Biology, Faculty of Medicine, Albert-Ludwigs University of Freiburg, Freiburg, Germany. [4] School of Computer Science and Engineering, Institute of Life Sciences, The Hebrew University of Jerusalem, Jerusalem, Israel. [5] Department of Biochemistry and Biophysics, Perelman School of Medicine, University of Pennsylvania, Philadelphia, PA, USA. ✉email: kenjim@pennmedicine.upenn.edu; fdaldal@sas.upenn.edu

Respiratory chains couple exergonic electron transport from nutrients to the terminal acceptor oxygen ($O_2$) and generate a proton motive force used for ATP synthesis. Complex I (NADH dehydrogenase) and Complex II (succinate dehydrogenase) are the entry points into the chain of reducing equivalents, reducing quinone (Q) to hydroquinone ($QH_2$). Complex III (cytochrome (cyt) $bc_1$ or $CIII_2$) oxidizes $QH_2$ to reduce cyt $c$, which in turn is oxidized by Complex IV (cyt $c$ oxidase or CIV) converting oxygen to water[1] (Fig. 1a).

Respiratory complexes are evolutionarily conserved, and bacterial enzymes are structurally simple, consisting mainly of the catalytic subunits. $CIII_2$ is a dimer with each monomer comprised of three subunits: the Rieske FeS (FeS) protein with a [2Fe-2S] cluster, cyt $b$ with hemes $b_H$ and $b_L$, and cyt $c_1$ with heme $c_1$ cofactors (Fig. 1a, b). The FeS protein external domain (FeS-ED) is mobile between the **b** (close to heme $b_L$) and **c** (close to heme $c_1$) positions[2,3]. Some bacterial species such as *Rhodobacter capsulatus*[4] and pathogens like *Helicobacter pylori*, *Campylobacter jejuni*[5], and *Neisseria*[6] contain only a high oxygen affinity $cbb_3$-type CIV to support their micro-aerophilic growth. The $cbb_3$-type CIV is a monomer comprised of four subunits: CcoN with heme $b$ and heme $b_3$-Cu binuclear center, CcoO with heme $c_o$, CcoQ, and CcoP with hemes $c_{p1}$ and $c_{p2}$ cofactors[7] (Fig. 1a, b).

Many Gram-negative bacteria contain soluble (e.g., cyt $c_2$) and membrane-anchored (via transmembrane domains or fatty acids) cyts $c$ (e.g., *R. capsulatus* cyt $c_y$[8], *Paracoccus denitrificans* cyt $c_{552}$[9], and *Helicobacter gestii* cyt $c_{553}$[10]) as electron carriers. Conversely, Gram-positive bacteria have no freely diffusing electron carrier but contain cyt $c$ domains fused to their $CIII_2$ like in *Mycobacterium smegmatis*[11], or CIV like in *Bacillus subtilis*[12]. In *R. capsulatus*, both the diffusible cyt $c_2$ and the membrane-anchored cyt $c_y$ carry electrons from $CIII_2$ to CIV in respiration[13].

In recent years, the co-occurrence of individual complexes together with multi-enzyme supercomplexes (SCs) in energy-transducing membranes has become evident[14–17]. SCs may enhance catalytic efficiency through substrate/product channeling or minimize production of harmful reactive oxygen species to decrease cellular distress[18]. The structures of mitochondrial SCs, such as $CICIII_2CIV$ or its smaller variants $CICIII_2$ and $CIII_2$-CIV[19] are well known[20,21]. Some bacterial SCs (e.g., *P. denitrificans*[22] or *C. glutamicum*[23]) have also been characterized biochemically, but only the structure of the Gram-positive *M. smegmatis* SC ($CIII_2CIV_2$) is available[24,25].

As of yet, no respiratory SC structure has been determined for Gram-negative bacteria, the evolutionary precursors of mitochondria. Furthermore, SCs containing $cbb_3$-type ancient forms of CIV with primordial respiratory features remain unknown[26]. Such structural studies have been hampered due to unstable interactions between $CIII_2$ and CIV, yielding trace amounts of "native" SCs. We have overcome this hurdle using a genetic approach and obtained large amounts of engineered SCs from the Gram-negative facultative phototroph *R. capsulatus*. Here we

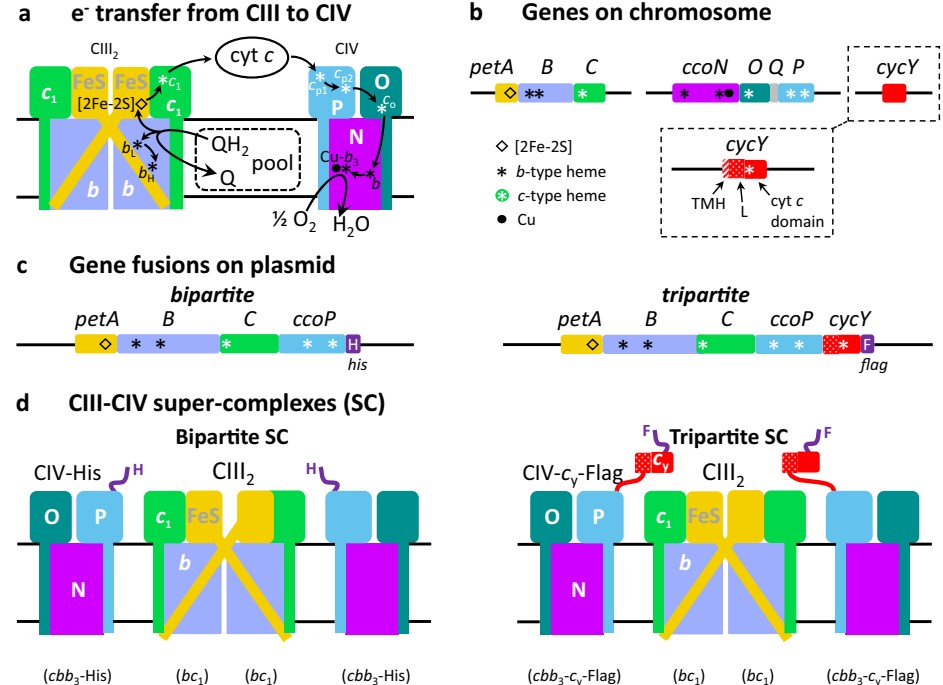

**Fig. 1 $QH_2$ oxidation and fused SCs. a** Bifurcated electron transfer reaction conveys one electron from $QH_2$ oxidation to the FeS protein (FeS, yellow) [2Fe-2S] cluster and another electron to cyt $b$ (periwinkle) hemes $b_L$ and $b_H$. Reduced [2Fe-2S] cluster transfers the electron to cyt $c_1$ (green) heme $c_1$ via the movement of the FeS protein from near heme $b_L$ to near heme $c_1$. The subsequent electron transfer step from heme $b_H$ to Q from the pool is not shown for clarity. A cyt $c$ ($c_2$ or $c_y$) carries electrons from heme $c_1$ to CIV, where electrons reach the heme-Cu (Cu-$b_3$) site, where $O_2$ is reduced to $H_2O$, via the CcoP (P, light blue) hemes $c_{p1}$ and $c_{p2}$, CcoO (O, dark green) heme $c_o$, and CcoN (N, purple) heme $b$. **b** *R. capsulatus petABC* encodes the structural genes of the $bc_1$-type $CIII_2$ subunits, the FeS protein (*petA*, yellow), cyt $b$ (*B*, periwinkle), and cyt $c_1$ (*C*, green), and *ccoNOQP* encodes those of the $cbb_3$-type CIV subunits, the CcoN (*ccoN*, purple), CcoO (*O*, dark green), CcoQ (*Q*, gray), and CcoP (*P*, light blue). *cycY* gene (red) encodes cyt $c_y$, and its 30-residue transmembrane helix (TMH), 69-residue linker (L), and 100-residue cyt $c$ (cyt $c$) domain are indicated. Heme cofactors of $b$- and $c$-type cyts are indicated by black and white asterisks, respectively, whereas diamond and dot designate the [2Fe-2S] cluster and Cu atom, respectively. **c** Plasmid-borne genetic fusions. The bipartite fusion (left) is formed by in-frame linking the 3′-end of *petC* to the 5′-end of *ccoP* and the tripartite fusion (right) by adding the linker and cyt $c$ domain of *cycY* to the 3′-end of *ccoP*. Colors and cofactors are as in **a**, and the his (H) and flag (F) affinity tags (dark purple) of the bipartite and tripartite fusion subunits are shown. **d** Schematic depiction of bipartite (left) and tripartite (right) supercomplexes (SC). The bipartite SC encodes a $CIII_2$ dimer fused on each side to a His-tagged CIV. The tripartite SC also contains the Flag-tagged cyt domain of $c_y$ (red) at the end of CcoP (blue).

report the cryo-electron microscopic (cryo-EM) structure of this functional $bc_1$-$cbb_3$-type SC (CIII$_2$CIV, at 5.2 Å resolution), together with three structural conformers of native CIII$_2$ (at 3.3–4.2 Å resolution). The structures show that the membrane-bound cyt $c_y$ and the diffusible cyt $c_2$ interact differently with CIV for electron transfer. This work illustrates the structural and functional features of a native-like CIII$_2$CIV and its distinct membrane-confined and membrane-peripheral respiratory electron transport pathways in Gram-negative bacteria.

## Results

### Isolation and characterization of engineered functional SCs.
Earlier studies have indicated that in *R. capsulatus* CIII$_2$, CIV, and cyt $c_y$ are in close proximity to each other[27]. Analyses of native membranes of this species indicated barely detectable entities around ~450 kDa $M_r$, larger than the masses of CIV monomers or CIII$_2$ dimers (Supplementary Fig. 1a, b). Similar entities were seen with dodecyl-maltoside or digitonin used to disperse wild-type membrane, but they were of low abundance and unstable, rendering their isolation and study difficult.

Earlier works of the assembly processes of CIII$_2$ and CIV showed that cyt $c_1$ interacts with cyt $b$ to form a cyt $b$-$c_1$ subcomplex[28], and similarly, CcoP associates with CcoNOQ subcomplex to yield an active CIV[29]. Thus we hypothesize that translationally fusing the C-terminus (C-ter) of cyt $c_1$ to the N-terminus (N-ter) of CcoP (on the inner ($n$) side of the membrane) as a cyt $c_1$-CcoP fusion might produce a bipartite $bc_1$-$cbb_3$-type SC that may form a native-like SC and enrich possible SC assembly factors during protein isolation for biochemical and structural studies (left panels of Fig. 1c, d). Further, adding the 69-residue linker (L) and the 100-residue cyt domain of $c_y$ to the C-ter of cyt $c_1$-CcoP (on the outer ($p$) side of the membrane) as a cyt $c_1$-CcoP-$c_y$ fusion might yield a tripartite SC with its attached electron carrier (right panels of Fig. 1c, d). This approach (see "Methods" and Supplementary Table 1) was successful and yielded high amounts of fused SCs of native-like electrophoretic mobilities (Supplementary Fig. 1a, b; a was overstained for IGA as compared to b). Indeed, these fused native-like SCs were functional in vivo and complemented a mutant lacking both CIII$_2$ and CIV for photosynthesis proficiency, which requires a functional CIII$_2$, and colonies producing them exhibited CIV activity in situ (Supplementary Fig. 1c). Thus the functional properties of the native-like SCs were like those of native SCs.

The His-tagged bipartite and Flag-tagged tripartite SCs were purified by tag-affinity and size exclusion chromatography (SEC; "Methods"; Supplementary Fig. 1d, e). Analyses of isolated proteins showed the presence of large entities of $M_r$ ~450 kDa range (Supplementary Fig. 1d, e fractions A-1 and B-1 in insets). These fractions contained the cyt $c_1$-CcoP (~65 kDa) or cyt $c_1$-CcoP-$c_y$ fusions (~80 kDa) (Supplementary Fig. 1f), and the protein bands were identified by mass spectrometry (MS) as subunits of CIII$_2$ and CIV (Supplementary Table 2). The fusion proteins contained covalently attached heme cofactor(s) as revealed by a cyt $c$-specific staining (Supplementary Fig. 1g). CcoQ ($M_r$ ~7 kDa) of CIV was absent in the SC preparations.

Purified SCs were characterized for their cyts $b$ and $c$ contents. The optical redox difference spectra were distinct from those of CIII$_2$[30] or CIV[31], and the tripartite SC contained more $c$-type cyt than the bipartite SC, due to the additional cyt domain of $c_y$ (Supplementary Fig. 1h). The samples had both QH$_2$:cyt $c$ reductase (specific to CIII$_2$) and cyt $c$ oxidase (specific to CIV) activities (Supplementary Fig. 1i, j). Thus the native-like SCs were enzymatically active in vitro. Moreover, the tripartite SC also exhibited QH$_2$:O$_2$ reductase (coupled CIII$_2$ + CIV) activity without addition of any electron carrier (e.g., horse heart cyt $c$ or *R.*

*capsulatus* cyt $c_2$; Supplementary Fig. 1k), suggesting that the cyt domain of $c_y$ fused to cyt $c_1$-CcoP transferred electrons from CIII$_2$ to CIV.

### Structures of the tripartite SCs.
Cryo-EM analysis of the tripartite SC preparations (Supplementary Fig. 1e, fraction B-1) showed that the initial three-dimensional (3D) classes were mainly of two different sizes (Supplementary Fig. 2, Box 1, left). The size (~180 Å length) and shape of the smaller particles suggested that these may correspond to a dimeric CIII$_2$ associated with a single CIV. Focused classification and processing of the subclass containing ~62,000 particles with the highest initial resolution and best discernable features yielded a tripartite CIII$_2$CIV map (SC-1A, EMD-22228) at 6.1 Å resolution (Supplementary Fig. 2a, "Methods"), while another dataset produced a map (SC-1B, EMD-22230) at 7.2 Å resolution (Supplementary Fig. 2b and Supplementary Table 3). The larger particles (~250 Å length, Supplementary Fig. 2, Box 1, left) represented a dimeric CIII$_2$ flanked by two CIV, as expected based on two $c_1$-CcoP-$c_y$ subunits per CIII$_2$. However, these particles were rare (~5000) and their map (SC-1C) could not be refined beyond ~10 Å resolution (Supplementary Fig. 2c).

A homology model of *R. capsulatus* $cbb_3$-type CIV was built using as a template the highly homologous *Pseudomonas stutzeri* structure (PDB: 3MK7; 3.2 Å resolution) ("Methods," Supplementary Table 3). The available CIII$_2$ model (PDB: 1ZRT; 3.5 Å resolution) was further refined (PDB: 6XI0; 3.3 Å resolution) based on the cryo-EM data obtained in this work (Supplementary Table 3). These models were fitted as rigid bodies into the maps SC-1A and SC-1B (Supplementary Fig. 3a), with correlation coefficients CC$_{box}$ of 0.75 and CC$_{box}$ of 0.71, respectively. The [2Fe-2S] clusters of the FeS proteins of CIII$_2$ could be recognized closer to **b** (heme $b_L$) than to **c** (heme $c_1$) positions but had lower occupancy and resolution likely due to conformational heterogeneity. The heterogeneity of the FeS-ED of CIII$_2$ in monomer B (away from CIV) was more pronounced than that in monomer A (adjacent to CIV) (Supplementary Fig. 3b). Lower resolutions of the CIII$_2$ FeS-ED portions were anticipated due to their mobility[2,3].

Superimposition of the CIII$_2$ portions of SC-1A and SC-1B maps showed that CIV was in different orientations in different maps (Supplementary Fig. 3c). The two extreme locations of CIV with respect to CIII$_2$ were displaced from each other by a translation of ~3 Å and a rotation of ~37° (Supplementary Fig. 3d, e; SC-1A in red and SC-1B in blue). Other subclasses identified in 3D classifications showed CIV in various orientations between those seen in SC-1A and SC-1B maps. This CIII$_2$CIV interface flexibility is attributed to the limited interaction between the CcoP (N-ter transmembrane helix (TMH)) of CIV and the cyt $b$ (TMH7) of CIII$_2$ (Fig. 2, top view). In the interface regions of SC-1A and SC-1B maps, additional weaker features that are not readily attributable to CIII$_2$ and CIV structures were observed. However, no membrane-external feature that might correspond to cyt domain of $c_y$, which is an integral part of the cyt $c_1$-CcoP-$c_y$ subunit, could be discerned in these maps.

### Structure of bipartite SC supplemented with cyt $c_y$.
The bipartite SC preparations (Supplementary Fig. 1d, fraction A-1) were supplemented with either purified full-length cyt $c_y$ or its soluble variant lacking the TMH (i.e., cyt S-$c_y$)[32] to yield the bipartite SC + $c_y$ and SC + S-$c_y$ samples. Following SEC, the elution fractions showed that only the intact cyt $c_y$, but not the cyt S-$c_y$, remained associated with the SC (Supplementary Fig. 4a). Thus the cyt domain of $c_y$ does not bind tightly to, and its TMH is required for its association with, the SC.

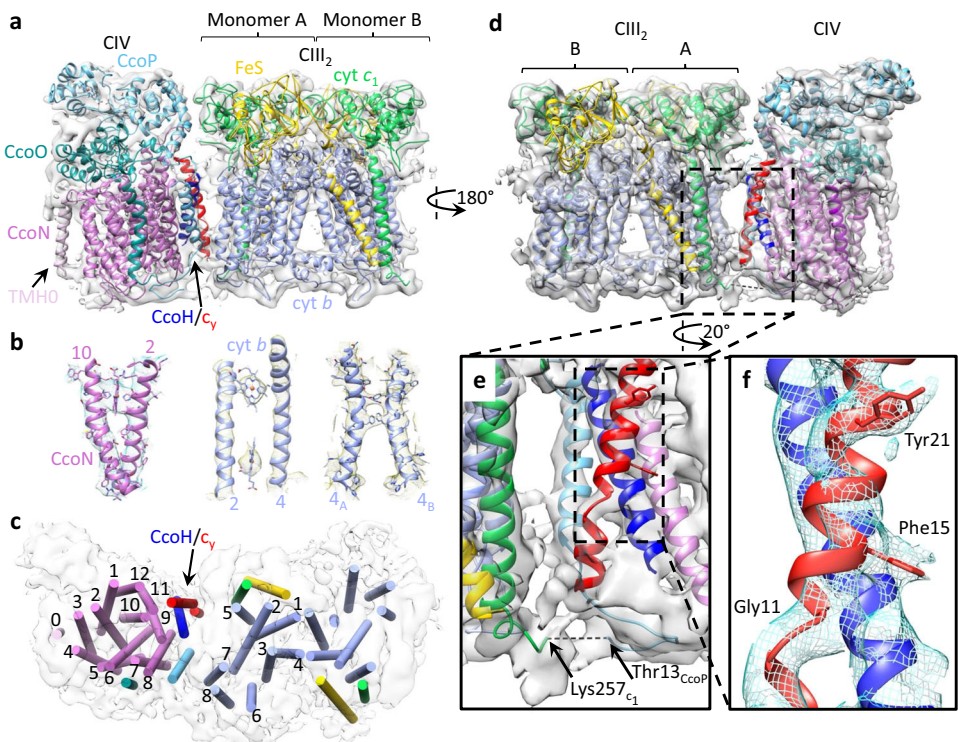

**Fig. 2 Cryo-EM structure of CIII₂CIV. a** The structures of CIII₂ (PDB: 6XI0) and the homology model of *R. capsulatus cbb₃*-type CIV were fitted into the cryo-EM map SC-2A (gray), depicted as a side view. All subunits are colored and labeled as in Fig. 1, and the arrow at the edge of CcoN of CIV is for the extra N-ter TMH (TMH0, light purple) specific to *R. capsulatus*. **b** Representative regions of the cryo-EM map showing map quality and model fitting. The TMH2 and TMH10 of CcoN show heme *b* and some bulky side chains. The protein backbone and hemes *b*L and *b*H are resolved between cyt *b* TMH2–TMH4 (compare to Fig. 3b, CIII₂ map at 3.3 Å). Large side chains are visible between the TMH4s of CIII₂ monomers A and B (4A and 4B). **c** Top view of CIII₂CIV with TMHs depicted as cylinders and colored as in **a**. The TMHs of cyt *b* (only CIII₂ monomer A) and CcoN of CIV are numbered, and the TMHs of the FeS protein (yellow), cyt *c₁* (green), CcoO (dark green), CcoP (light blue), CcoH/*c*y (blue/red with an arrow), and CcoN TMH0 (light purple) are shown. **d** 180° rotated view for the back view of CIII₂CIV interface. The two extra TMHs at the interface are those of CcoH (blue) and cyt *c*y (red). **e** Enlarged view of CIII₂CIV interface. The view is slightly rotated relative to **d** for better visibility of CcoP TMH in the background (light blue). CcoN TMH9 is next to CcoH and cyt *c*y TMHs. The fusion region between cyt *c₁* and CcoP is indicated at the bottom, with cyt *c₁* C-ter (green) and CcoP N-ter (light blue) with their respective end residues (Lys257$_{c1}$ and Thr13$_{CcoP}$) (only resolved portions). The 12 N-ter CcoP residues connecting these two chains (dashed line) are not resolved. **f** Enlarged view showing close interaction between CcoH and cyt *c*y TMHs. Characteristic features of cyt *c*y TMH (NH₂-Gly11xxxPhe15xxxxxTyr21-COOH) are used to determine the registration, with the helix break induced by Gly11, and bulky side chains for Phe15 and Tyr21 clearly visible.

The cryo-EM analyses of the bipartite SC + *c*y yielded a map (SC-2A, EMD-22227) at 5.2 Å resolution (Supplementary Fig. 5a, b), with local resolutions ranging from 4.3 to 8.0 Å (Supplementary Fig. 6a, c). The homology model of CIV and the refined model of CIII₂ (PDB: 6XI0) were fitted as rigid bodies into SC-2A (Fig. 2a), with a correlation coefficient CC$_{box}$ of 0.74. Comparison of the maps SC-2A (bipartite CIII₂CIV) with SC-1A (tripartite CIII₂CIV) showed that they were similar, with root-mean-square deviation of 1.6 Å. Hence, they are collectively referred to as CIII₂CIV, irrespective of their bipartite or tripartite origins.

The dimensions (~155 × 60 × 90 Å, $L × W × H$) of the slightly curved CIII₂CIV structure were consistent with a CIII₂ dimer associated with one CIV. On SC-2A map (5.2 Å resolution), some large aromatic side chains could be readily discerned (Fig. 2b). Of the TMHs seen, 34 were accounted for by two FeS proteins, two cyts *b*, and two cyts *c₁* (2, 16, and 2 TMHs per dimer, respectively) and single CcoN, CcoO, and CcoP (12, 1, and 1 TMHs, respectively) (Fig. 2c). The features corresponding to the heme cofactors of CIII₂CIV were attributed to hemes *b*H and *b*L of cyt *b*, heme *c₁* of cyt *c₁*, and to hemes *b* and *b₃* of CcoN, heme *c*o of CcoO, and hemes *c*p1 and *c*p2 of CcoP proteins. Like the tripartite maps, in this case also the FeS-EDs of CIII₂ had lower resolution and the [2Fe-2S] clusters were located closer to heme

$b$L (**b** position). An additional TMH was observed at the distal end of CIV (Fig. 2a, rotated 180° in Fig. 2d) close to CcoN TMH3 and TMH4 (Fig. 2c). Due to its location, this TMH (depicted in Fig. 2 as an ab initio model of CcoN residues Arg25 to Leu48) was tentatively attributed to the extra N-ter TMH (TMH0) of CcoN, specific to *R. capsulatus*.

The interface of CIII₂CIV is delimited by CcoN TMH8 and TMH9, CcoP TMH, cyt *b* TMH5 and TMH7, and cyt *c₁* TMH of monomer A, with the closest interaction being between CcoP TMH and cyt *b* TMH7 (Fig. 2a, c). Two interacting TMHs of unknown identities (Fig. 2d, in red and blue) and an inter-complex connection are seen at the interface. This connection linking CIII₂ to CIV is at the *n* face of the membrane, near the TMHs of cyt *c₁* and CcoP (Fig. 2e, Lys257$_{c1}$ and Thr13$_{CcoP}$). It is tentatively attributed to the 13 amino acid residues spanning the fusion joint between the C-end of PetC and N-end of CcoP (with no extra amino acid inserted in between) that are not clearly observed.

## Assembly factor CcoH and cyt *c*y TMHs are located at CIII₂-CIV interface. A co-evolution-based approach predicting the residue–residue contacts in protein–protein interactions,

RaptorX-ComplexContact[33], was used to identify the TMHs at $CIII_2CIV$ interface. The single TMH containing CIV-related proteins (i.e., CcoQ subunit, CcoS and CcoH assembly factors[29], and cyt $c_y$[8]) were analyzed against the subunits of $CIII_2$ and CIV. Significant predictions of interacting residue pairs (confidence value >0.5) were found only between CcoN (primarily TMH9) and the putative N-term TMH of CcoH (Supplementary Fig. 7a, b). An ab initio model of CcoH TMH was docked via Patch-Dock[34] onto CIV using the predicted residue–residue contacts as distance restraints (15 Å threshold; "Methods"). The top scoring 50 models converged to a single cluster at the location of the unknown TMH, next to CcoN TMH9 (Supplementary Fig. 7c). Examination of the interactions between CcoH TMH and CcoN TMH9 showed multiple co-evolutionarily conserved residues in close contacts (Supplementary Fig. 7b, d). Earlier studies had indicated that CcoH can be cross-linked by disuccinimidyl suberate (spacer length ~11 Å) to CcoP and CcoN[35]. Thus the TMH close to CcoN TMH9 (Fig. 2e, blue TMH) is tentatively assigned to the assembly factor CcoH.

A clear difference between the maps SC-2A (bipartite $CIII_2CIV + c_y$) and SC-1A (tripartite $CIII_2CIV$) was seen at the interface. The unidentified TMHs were barely visible in SC-1A but highly enhanced in SC-2A (Fig. 2e), indicating higher occupancy. The observation that only the native cyt $c_y$ binds to bipartite SC via its TMH suggested that the TMH (red in Fig. 2e) next to CcoH TMH (blue in Fig. 2e) corresponds to the membrane anchor of cyt $c_y$. This explanation is supported by fact that the bipartite $CIII_2CIV + c_y$ is supplemented with full-length cyt $c_y$ while the tripartite SC contains only the fused cyt domain of $c_y$. Indeed, landmark densities corresponding to the helix-breaking Gly11 and the correctly spaced bulky side chains of Phe15 and Tyr21 of cyt $c_y$ TMH ($NH_2$-xxxGly11xxxPhe15xxxxx-Tyr21-COOH) were discerned (Fig. 2f).

We noted that some $CIII_2CIV + c_y$ subclasses exhibited a weak feature on the $p$ side of the membrane that may reflect the cyt domain of $c_y$ (Supplementary Fig. 5g, SC-2B). However, this feature could not be refined to high resolution, consistent with the weak binding of cyt domain of $c_y$ to $CIII_2CIV$. Moreover, upon addition of cyt $c_y$, the predominant conformation of CIV in the bipartite $CIII_2CIV + c_y$ (Supplementary Fig. 5a, b, SC-2A) shifted toward that seen in map SC-1A of tripartite SC (Supplementary Fig. 2a), with no major class corresponding to SC-1B (Supplementary Fig. 2b). This suggested that local interactions between the CcoH and cyt $c_y$ TMHs and CIV decreased the interface flexibility of $CIII_2CIV$ (Fig. 2e).

**Cryo-EM structures of *R. capsulatus* native $CIII_2$.** The bipartite SC + $c_y$ samples also contained smaller particles (~110 Å length) (Supplementary Fig. 2, Box 2) that were the size of $CIII_2$ Supplementary (Fig. 5c, d). Analyses of these particles using C2 symmetry led to the map $CIII_2$ at 3.3 Å resolution for native $CIII_2$ (Supplementary Fig. 5e), with local resolutions ranging from 3.0 to 4.0 Å (Supplementary Fig. 6b, d and Supplementary Table 3). The FeS-ED parts showed lower occupancy and resolution compared to the rest of the map, indicating conformational heterogeneity. When similar analyses were carried out without imposing C2 symmetry, three distinct maps were obtained for $CIII_2$ ($CIII_2$ c-c, $CIII_2$ b-c, and $CIII_2$ b-b at 3.8, 4.2, and 3.5 Å resolutions, respectively; Supplementary Fig. 5f). These maps were superimposable with respect to cyt $b$ and cyt $c_1$ subunits, except for the FeS-ED portions. The $CIII_2$ structures depicted by the $CIII_2$ b-b (Fig. 3a–c) and $CIII_2$ c-c (Fig. 3d) maps exhibited overall C2 symmetry, but in the former the FeS-EDs were located in **b**, whereas in the latter in **c** position[3]. Notably, the third structure ($CIII_2$ b-c) was asymmetric, with the FeS-ED of one

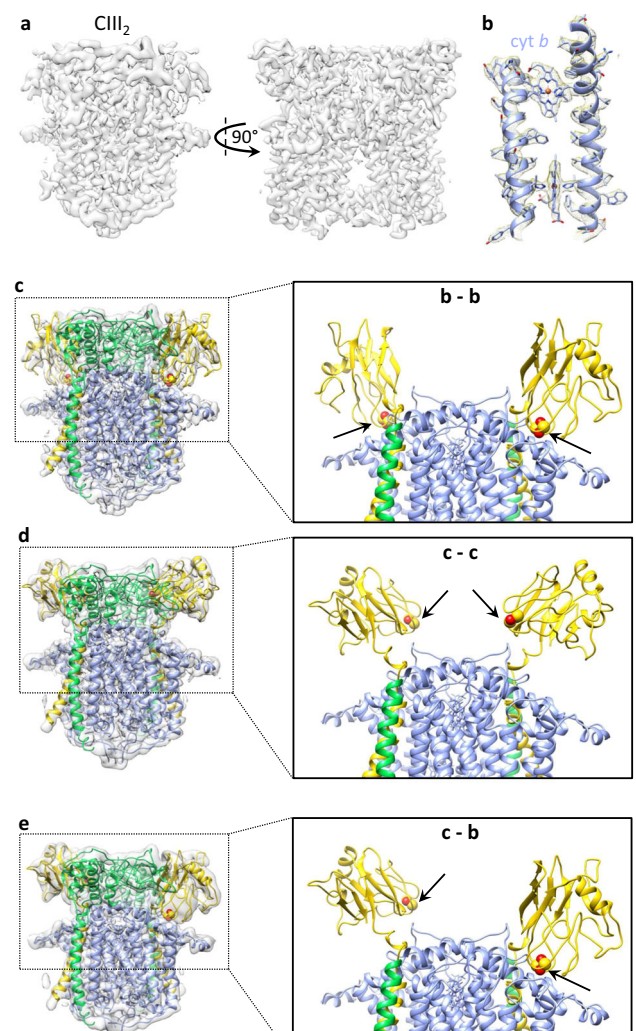

**Fig. 3 Structures of native $CIII_2$ conformers. a** Cryo-EM map $CIII_2$ b-b (EMD-22226) with both FeS proteins in **b** position. **b** Representative region of $CIII_2$ b-b demonstrating map quality and model fitting. TMH2 and TMH4 of cyt $b$ with hemes $b_L$ and $b_H$ are shown. **c–e** Maps and models showing different conformations of the FeS proteins. The left panels show the $CIII_2$ structure fitted into the different maps with the subunit colorings (cyt $b$ in periwinkle, cyt $c_1$ in green, and the FeS protein in yellow) as in Fig. 2. The right panels show the top half of the models with the membrane-external domain of cyt $c_1$ omitted to better visualize the b-b, c-c, and c-b positions of FeS-EDs. The [2Fe-2S] clusters are shown as yellow–red spheres and indicated by arrows. **c** Structure of native $CIII_2$ with both FeS-EDs in **b** position (map $CIII_2$ b-b, EMD-22226; PDB: 6XKV). **d** Structure of native $CIII_2$ with both FeS-EDs in **c** position (map $CIII_2$ c-c, EMD-22224; PDB: 6XKT). **e** Structure of native $CIII_2$ with one FeS-ED in **c** and one in **b** position (map $CIII_2$ b-c, EMD-22225; PDB: 6XKU).

monomer being in **c** and the other one in **b** positions (Fig. 3e). Such asymmetric structures of native $CIII_2$ have not been observed frequently, although they are proposed to occur during $QH_2$ oxidation[36–38].

**Interactions of cyt $c_2$ and cyt $c_y$ with $CIII_2CIV$.** As the tripartite or bipartite SC maps contained no feature that could clearly represent cyt domain of $c_y$, the interaction interfaces between SC and its electron carrier remained unknown. Thus a docking approach combined with crosslinking mass spectrometry

(XL-MS) was used to characterize these protein-protein interactions. XL-MS has been used successfully in vivo with whole cells[39–42] or in vitro with purified materials[43,44], and PatchDock[34] is an efficient rigid docking method that maximizes geometric shape complementarity. First, the co-crystal structure (PDB: 3CX5) of yeast $CIII_2$ with its electron carrier iso-1 cyt $c$[45] was used as a template (homology between yeast cyt $c_1$ and *R. capsulatus* cyt $c_1$: 31% identity and 58% similarity; iso-cyt $c$ and cyt $c_2$: 25% identity and 56% similarity) to model the binding of cyt $c_2$ on bacterial $CIII_2$. The *R. capsulatus* cyt $c_1$ (PDB: 6XI0) and cyt $c_2$ (PDB: 1C2N) structures were superimposed with their yeast counterparts on the co-crystal structure, and a model with a single cyt $c_2$ docked to a monomer of $CIII_2$ was generated (Supplementary Fig. 4e). This docking model was then experimentally validated using the protein crosslinker 4-(4,6-dimethoxy-1,3,5-triazin-2-yl)-4-methyl-morpholinium chloride (DMTMM). The crosslinks (XLs) were identified using MeroX[44,46] search engine, and among them those of high confidence were cross-validated using FindXL[43,47]. The intra-subunit XLs within $CIII_2CIV$ provided controls (Supplementary Fig. 4c, d, "Methods"). The three XLs between cyt $c_1$ and cyt $c_2$ supplied distance restraints for docking cyt $c_2$ to $CIII_2$ using PatchDock (Supplementary Table 4 and Supplementary Fig. 4f), and the docking models (top 50) clustered at a single region per monomer of $CIII_2$ (Fig. 4a, right). This region overlapped with the binding site of cyt $c_2$ defined by the model obtained by alignment to the yeast co-crystal structure (Supplementary Fig. 4g). The distance from cyt $c_2$ heme-Fe to cyt $c_1$ heme-Fe was ~16.8 Å for the co-crystal-derived model, while comparable distances were between ~13.8 and 20.4 Å with the docking models. Thus docking via PatchDock integrating XL-MS-mediated distance restraints defined reliably, but with limited accuracy, the interaction region of cyt $c_2$ on $CIII_2$.

No information about the binding of cyt $c_2$ and $cbb_3$-type CIV was previously available. First, this binding was shown biochemically (Supplementary Fig. 4b) and then subjected to XL-MS using DMTMM. The XLs between the proteins (1 between cyt $c_2$ and CcoP and 8 between cyt $c_2$ and CcoO) provided distance restraints for docking cyt $c_2$ to CIV (Supplementary Table 4). The docking models (top 50) clustered in a single region of CIV (Fig. 4a, left), closer to heme $c_{p2}$ (cyt $c_2$ heme-Fe to $c_{p2}$ heme-Fe: ~15.2–35.6 Å) than heme $c_{p1}$ (cyt $c_2$ heme-Fe to $c_{p1}$ heme-Fe: ~23.0–42.0 Å) of CcoP (Fig. 5). The two cyt $c_2$-binding regions on $CIII_2CIV$ are highly distant from each other (closest $c_2$ heme-Fe on $CIII_2$ to that on CIV is ~69 Å; Fig. 5a).

Next, the binding interactions between cyt domain of $c_y$ and $CIII_2CIV$ were sought using the longer, primary amine reactive disuccinimidyl dibutyric urea (DSBU) together with the shorter, carboxylic acid reactive DMTMM. The use of two crosslinkers with different chemical properties and spacer lengths provided better coverage and tighter distance restraints, as the SC interface and the FeS-ED of $CIII_2$ were, and the cyt domain of $c_y$ was expected to be, flexible (see below). Similar to DMTMM, DSBU yielded multiple intra-subunit XLs within the subunits of $CIII_2CIV$ as experimental controls (Supplementary Table 4 and Supplementary Fig. 4c, d). Six XLs (five cyt $c_y$ to cyt $c_1$ and one cyt $c_y$ to FeS protein) with DMTMM and four XLs (only cyt $c_y$ to FeS protein) with DSBU were found (Supplementary Table 4). No XL was observed between cyt $c_y$ and CIV, suggesting that cyt $c$ domain of $c_y$ is closer to $CIII_2$ in $CIII_2CIV$. In this case, the models (top 50) generated by PatchDock clustered in two binding regions for cyt domain of $c_y$ on each $CIII_2$ monomer of SC. One of the clusters was on top of cyt $c_1$ and overlapped with cyt $c_2$ cluster (Fig. 4b), whereas the other one was between cyt $c_1$ and the FeS-ED near $CIII_2$ inter-monomer space (Supplementary Fig. 4h–j). These two clusters are best visible in a top view

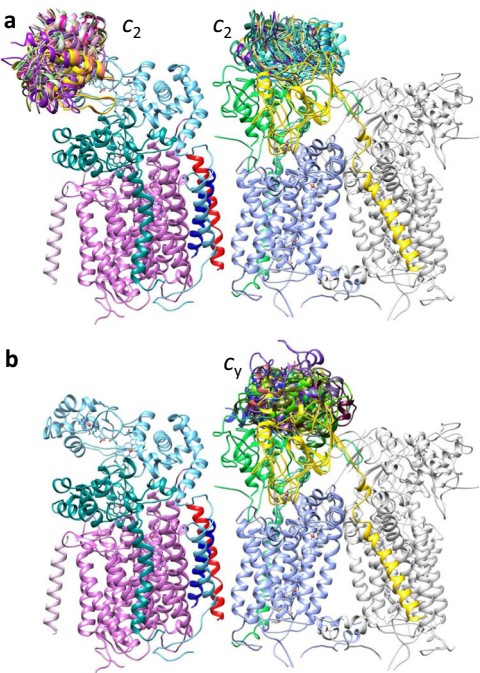

**Fig. 4 Binding regions of cyt $c_2$ and cyt $c_y$ on $CIII_2CIV$.** The binding regions were defined by XL-MS-guided docking, and the subunits of $CIII_2CIV$ are colored as in Fig. 2 except the monomer B of $CIII_2$ shown in light gray. Only the binding regions on monomer A are shown. **a** Cyt $c_2$ (PDB: 1C2N) was docked onto $CIII_2$ and CIV using PatchDock guided by DMTMM-generated XLs as distance restraints. This yielded one cluster of models on CIV and one per monomer of $CIII_2$. **b** A model of cyt domain of $c_y$, generated using *P. denitrificans* cyt $c_{552}$ structure (PDB: 3M97) as a template (RMSD between template and model: 0.2 Å) was docked on $CIII_2$ as in **a**, except that both DMTMM- and DSBU-generated XLs provided distance restraints. Two binding clusters for cyt domain of $c_y$ per monomer of $CIII_2$ were found. The two clusters are located behind each other on a side view but are clearly visible on top views (Supplementary Fig. 4h–j, labeled 1 and 2). Only cluster 1 that is closer to cyt $c_1$ and overlapping with the binding region of cyt $c_2$ is shown. In all cases, ten representative models are shown to visualize the clusters of binding models. No binding region for cyt $c_y$ on CIV could be defined since no XL was found between the proteins.

(Supplementary Fig. 4j), and the distances between cyt $c_y$ heme-Fe and cyt $c_1$ heme-Fe of $CIII_2$ monomer A are between 13.8 and 47.1 Å (Fig. 5). To further support these binding locations, we searched in our cryo-EM datasets 3D classes that contained extra features attributable to cyt domain of $c_y$, and a minor class (~18,000 particles; Supplementary Fig. 5g) showing such an extra feature between CIV and CIII was found (Supplementary Fig. 4h, i).

To better define the clusters of cyt domain of $c_y$, the docking analyses were repeated using the three different conformers of native $CIII_2$ (Fig. 3c–e, $CIII_2$ b-b, c-c, and b-c). Interestingly, when the FeS-EDs are in **c** position ($CIII_2$ c-c), the docking models gathered as a single cluster on cyt $c_1$, slightly displaced toward the FeS-ED of the same monomer (Supplementary Fig. 8a–c). However, when the FeS-EDs are in **b** position ($CIII_2$ b-b), such models were more spread out (Supplementary Fig. 8d–f). The third model with one FeS-ED in **c** and the other in **b** positions yielded the expected clustering pattern depending on the local FeS-ED conformation. In $CIII_2CIV$ both FeS-EDs being in the **b** position, we assumed that cyt domain of $c_y$ docks similarly to that seen with $CIII_2$ b-b and that spreading of docking originates from the variable conformations of the FeS-EDs in SC (Supplementary Fig. 8). Heme $c_1$ and not the FeS protein being

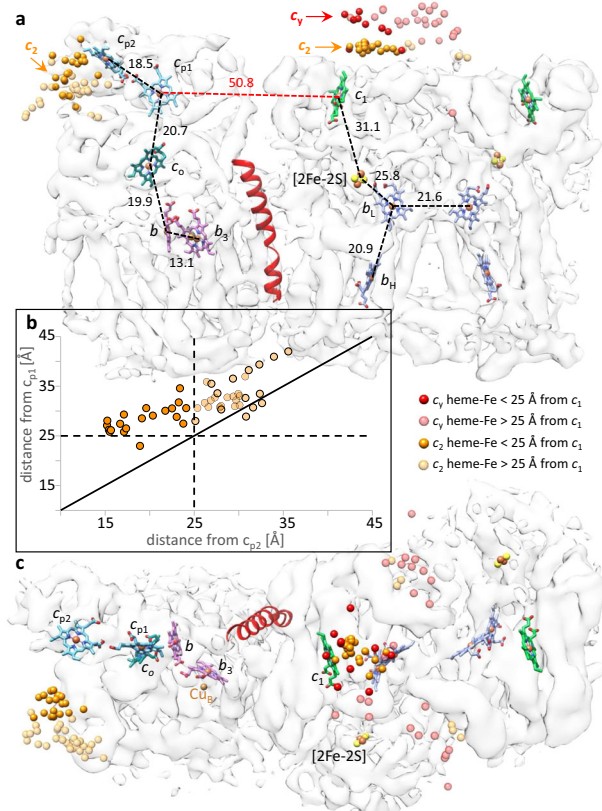

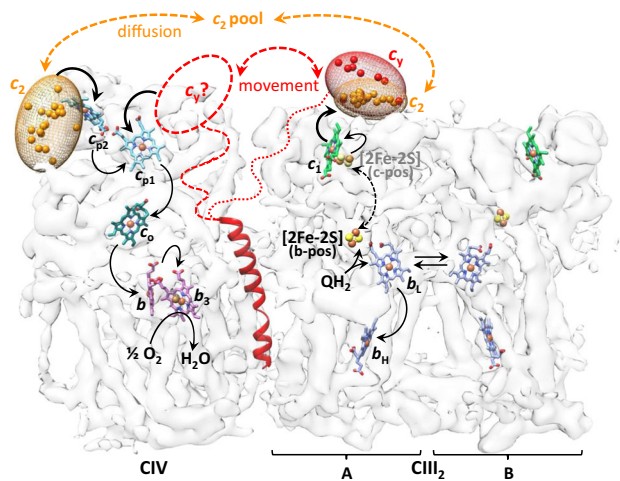

**Fig. 5 Organization of CIII₂CIV cofactors and redox partners. a** The cofactors are shown inside the transparent map SC-2A of CIII₂CIV with the same subunit colors as in Fig. 2: hemes $b_L$ and $b_H$ (periwinkle), heme $c_1$ (green), hemes $c_{p1}$ and $c_{p2}$ (light blue), heme $c_o$ (dark green), hemes $b$ and $b_3$ (purple). The [2Fe-2S] clusters are shown as yellow–red spheres. In all cases, the distances (heme-Fe to heme-Fe) between the heme cofactors are indicated. The positions of docked cyt $c_2$ and cyt domain of $c_y$ are indicated as orange (heme $c_2$) and red (heme $c_y$) spheres, respectively, representing their heme-Fe atoms. All heme-Fe atoms corresponding to the top 50 docking positions for cyt $c_2$ on CIV are shown as solid (<25 Å) or transparent (>25 Å) spheres, depending on their distances to heme $c_{p2}$. In the case of CIII₂, only the docking positions of cyt $c_2$ and cyt $c_y$ on monomer A and between the monomers A and B are shown, omitting those located on monomer B. The TMH of cyt $c_y$ is shown in red at CIII₂CIV interface. **b** The heme-Fe atoms of all 50 cyt $c_2$ models docked onto CIV are plotted as function of their distances from heme $c_{p1}$ and heme $c_{p2}$, with the Fe atoms within 25 Å shown as solid spheres and those beyond 25 Å as transparent spheres. The vast majority of heme-Fe atoms of docked cyt $c_2$ models are closer to heme $c_{p2}$ than heme $c_{p1}$ of CIV. **c** Top view of the map shown in **a** is presented to better visualize the distribution of the docked cyt domain of cyt $c_y$ on monomer A and between the monomers A and B. In all cases, the heme-Fe atoms are depicted by spheres and colored as indicated above and on the figure.

**Fig. 6 Proposed cyt $c_2$ and cyt $c_y$ binding regions of CIII₂CIV and electron transfer pathways.** The likely binding regions of cyt $c_2$ and cyt $c$ domains of $c_y$ (orange and red ellipsoids, respectively), defined by XL-MS-guided docking, are depicted by the distributions of their heme-Fe atoms on the transparent map SC-2A of CIII₂CIV. Only the positions that are within 25 Å of heme $c_1$ of CIII₂ or heme $c_{p2}$ of CIV are indicated. The cofactors together with the TMH of cyt $c_y$ are shown as in Fig. 5. The linker region (indicated by dotted or dashed lines) between the TMH and the cyt domain of $c_y$ is not resolved in the cryo-EM map. The proposed electron transport pathways are shown by thicker black arrows: upon QH₂ oxidation by CIII₂, cyt $c_y$ that is integral to CIII₂CIV receives an electron from heme $c_1$. It then moves (double-headed dashed red arrow) to an undefined binding region (dashed oval with $c_y$?) on CIV, where it delivers the electron to the nearest heme $c_{p1}$ of CIV. Similarly, cyt $c_2$ that is peripheral to CIII₂CIV also receives an electron from heme $c_1$, diffuses away to reach CIV, and conveys it to heme $c_{p2}$. The canonical electron transfers occurring from QH₂ to heme $c_1$ in CIII₂, and from heme $c_{p1}$ to O₂ in CIV, are indicated by thinner arrows. The double headed dashed black arrow depicts the movement of the [2Fe-2S] of FeS protein from the **b** position (b-pos, black, solid spheres) to the **c** position (c-pos, gray, transparent spheres) in CIII₂ during QH₂ oxidation. A possible electron equilibration between the two heme $b_L$ of CIII₂ is indicated by double arrows, and the electron transfer steps subsequent to heme $b_H$ reduction are not shown.

frequent XLs to CIII₂ led us to conclude that cyt $c_y$ oscillates within CIII₂CIV during QH₂:O₂ oxidoreduction (Fig. 6).

## Discussion

Prior to this work, no information was available neither on the occurrence or structure of any $cbb_3$-type CIV containing SC nor on its interactions with physiological electron carriers. Here we describe the cryo-EM structures of engineered native-like active $bc_1$-$cbb_3$-type respiratory SCs, together with three structural conformers of native CIII₂ at high resolution, from the Gram-negative, facultative phototroph *R. capsulatus*. Although the 3D structures of bacterial $bc_1$-type CIII₂ are available, native CIII₂ heterodimers have not been reported. Similarly, only one structure was available for $cbb_3$-type CIV[50]. Members of this subfamily of heme-Cu:O₂ reductases are widespread among bacteria and essential for microaerobic processes, including anaerobic photosynthesis and bacterial infection[4].

The facultative CIII₂CIV of *R. capsulatus* is more flexible, possibly due to the involvement of CIII₂ into multiple metabolic pathways, unlike the highly rigid obligate CIII₂CIV₂ SC of *Actinobacteria* dedicated to respiration[24,25]. In species like *R. capsulatus*, the dual function of CIII₂, interacting with the photochemical reaction center during photosynthesis and with

the electron exit site of CIII₂[48,49], the cluster on cyt $c_1$ was taken as the productive binding region of cyt domain of $c_y$.

Examination of all pertinent distances between the cofactors of CIII₂CIV (Fig. 5a) indicates that the binding region of cyt domain of $c_y$ near heme $c_1$ is far from CIV. The large distance (~50.8 Å) separating cyt $c_1$ heme-Fe of CIII₂ monomer A from CcoP $c_{p1}$ heme-Fe (the closest compared with heme $c_{p2}$) makes it impossible to define a position for cyt $c_y$ close enough to heme $c_1$ reducing it, and heme $c_{p1}$ oxidizing it, while transferring electrons from CIII₂ to CIV. The distance constraint, the inability to resolve the cyt domain of cyt $c_y$ by cryo-EM, and the more

CIV during respiration, may need this natural flexibility to allow metabolic adaptations for growth fitness. The native-like SCs, obtained by genetically fusing a subunit of $CIII_2$ to another one of CIV, are fully active both in vivo and in vitro and exhibit increased stability. These SCs are produced in cells without any harmful effect, mature properly, and support growth of mutants lacking either $CIII_2$ or CIV or both. Moreover, detergent-dispersed membranes and affinity-purified SCs exhibit both $CIII_2$ and CIV, as well as $CIII_2$-CIV coupled, enzymatic activities. Clearly this innovative approach, although yielded heterogeneous preparations, containing mixtures of $CIII_2CIV_2$, $CIII_2CIV$, and $CIII_2$ particles, allowed resolution of their structures and defined the likely binding regions of cyt $c_2$ and cyt $c_y$. Insertion of different spacers at the cyt $c_1$-CcoP fusion junction and over-expression of the subunits and related assembly components could not further improve this heterogeneity ("Methods"). Certainly, it remains unknown to what extent the structural and functional properties of the native-like SCs described here are identical to their "native" counterparts, which are currently unavailable.

The structures of the tripartite $CIII_2CIV$, which contains a fused cyt $c$ domain[39], and the bipartite $CIII_2CIV + c_y$, which has a bound native cyt $c_y$, were highly similar at sub-nanometer resolution (~7.2–5.2 Å, respectively). Their flexibility was due to limited protein–protein interactions at their interface, where the TMHs of cyt $c_y$ and CcoH are located (Fig. 2). Previously, neither the exact location nor the mobility of cyt $c_y$ between $CIII_2$ and CIV were known. The SC structure shows that the N-terminal TMH of cyt $c_y$ is locked at the interface, allowing mobility of its cyt domain within $CIII_2CIV$ (Fig. 6). The linker region of cyt $c_y$ remains unresolved but is long enough to reach both $CIII_2$ and CIV. Early studies with R. capsulatus cyt $c_y$ had shown that a full-length linker (69 residues) is needed for rapid (~<50 μs) electron transfer from $CIII_2$ to the photosynthetic reaction center[51], whereas a shorter linker (~45 residues) is proficient for respiratory electron transfer to CIV[52].

In native $CIII_2$ conformers, different positions of the [2Fe-2S] cluster bearing FeS-EDs were observed. Crystallographic structures have often depicted bacterial $CIII_2$ as symmetrical homodimers[3,53,54]. These structures were obtained in the presence of inhibitors constraining FeS-EDs near heme $b_L$ or used mutants stabilizing it on cyt $b$ surface. Alternatively, they contained crystal contacts restricting the FeS-ED movement[55]. To our knowledge, no native heterodimeric $CIII_2$ structure of bacterial origin with different conformations of its FeS-EDs has been reported. Only recent cryo-EM structures of mitochondrial SCs have shown different maps for $CIII_2$ FeS-EDs[17,56]. Thus native $CIII_2$ is not always a symmetric homodimer, and the FeS-ED of each monomer is free to move independently from the other. The Q-cycle models describe the mechanism of $CIII_2$ catalysis by two turnovers of a given monomer, with the different positions of the FeS-ED protein attributed to different catalytic steps[3,48,49]. Emerging asymmetric structures of bacterial and mitochondrial native $CIII_2$ obtained by cryo-EM in the absence of inhibitors or mutations, combined with the inter-monomer electron transfer between the hemes $b_L$ of the monomers[57,58], now provide a glimpse into plausible "heterodimeric Q-cycle" mechanism(s)[37,38]. Accordingly, the FeS-EDs of $CIII_2$ monomers might cycle between homo- and hetero-dimeric conformations during the Q-cycle.

This work defined hitherto unknown structural interactions between the cyt $c_2$ or cyt $c_y$ and native-like $CIII_2CIV$ (Fig. 6). The structures indicate that the distances separating heme $c_1$ of $CIII_2$ monomer A and hemes $c_{p1}$ or $c_{p2}$ of CIV are too large (Fig. 5) for direct microsecond time scale electronic communication[59] to sustain the turnover rate of $CIII_2CIV$. Thus, even when $CIII_2$ and

CIV form a SC, an electron carrier is required for $QH_2$:$O_2$ oxidoreduction. The binding location of cyt $c_2$ on CIV defined here, the redox midpoint potentials ($E_m$) of the cofactors, and the distances separating them (Fig. 5a) suggest that cyt $c_2$ would confer electrons to the closer heme $c_{p2}$, rather than the more distant heme $c_{p1}$, of CcoP. This will then initiate canonical electron transfer via heme $c_{p1}$, heme $c_o$, and heme $b$ to heme $b_3$-$Cu_B$ site for $O_2$ reduction[60,61] (Fig. 6). For purified R. capsulatus proteins, the $E_m$ values of cyt $c_2$, cyt $c_1$[30], and heme $c_o$ of CcoO are known[31]. The $E_m$ values of R. capsulatus CcoP hemes $c_{p1}$ and $c_{p2}$ are unknown, but based on similar $E_m$ values of heme $c_o$ for Bradyrhizobium japonicum and R. capsulatus, they are expected to be similar[62].

The likely interaction region of cyt $c_y$ on cyt $c_1$ of $CIII_2$, which is close to that of cyt $c_2$, is now known but that on CIV remains less well defined. The distance separating the redox centers is a major factor that controls the rate of electron transfer[59]. The binding region of cyt domain of $c_y$ on $CIII_2$ suggests that reduced cyt $c_y$, upon its movement to CIV, might preferentially convey electrons to the closer heme $c_{p1}$ than heme $c_{p2}$ of CcoP (Fig. 6). If so, under physiological conditions, heme $c_{p1}$ would be the primary receiver of electrons derived from $QH_2$ oxidation by $CIII_2$, forming a fully membrane-confined electronic wiring within $CIII_2CIV$. In contrast, cyt $c_2$ carries electrons from heme $c_1$ to heme $c_{p2}$ via free diffusion. This membrane-external pathway might accommodate electrons not only from $QH_2$ but also from other donors distinct from $CIII_2$. As such, reduction of cyt $c_2$ during methylamine oxidation[63] or degradation of sulfur containing amino acids, converting toxic sulfite ($SO_3^{2-}$) to sulfate ($SO_4^{2-}$) by sulfate oxidase[64], might provide electrons to CIV, contributing to energy production.

In summary, the architecture of an engineered native-like $CIII_2CIV$ along with its interactions with its physiological redox partners establishes salient structural and functional features of two distinct respiratory electron transport pathways (membrane-confined and membrane-external) that operate between $CIII_2$ and CIV in Gram-negative bacteria.

## Methods

**Bacterial strains, growth conditions, and primers used**. Bacterial strains, plasmids, and primers used are listed in Supplementary Table 1. LB medium supplemented as appropriate with ampicillin, gentamicin, kanamycin, or tetracycline at 100, 12, 50, or 12.5 μg/mL, respectively, was used for growing Escherichia coli strains at 37 °C. R. capsulatus strains were grown chemoheterotrophically under semi-aerobic/dark conditions at 35 °C on enriched (MPYE) medium, supplemented as needed with gentamicin, kanamycin, spectinomycin, or tetracycline at 3, 10, 10, or 2.5 μg/mL, respectively. Colonies were stained for $cbb_3$-type CIV activity by incubating plates with a 1:1 (v/v) mixture of 35 mM 1-naphtol and 30 mM N,N-dimethyl-1,4-phenylenediamine (NADI staining)[65].

**Construction of $petABC::ccoP$-His$_8$ (cyt $bc_1$-CcoP) fusion**. Using the primers Fw-ccoP (StuI) and Rv-ccoP (HindIII) (Supplementary Table 1), $petABC::ccoP$-His$_8$ fusion was constructed by ligating in-frame the PCR-amplified 1.13 kb StuI-HindIII fragment containing $ccoP$ (with a C-terminal His$_8$ tag) to the 3′ end of $petABC$, after elimination of $petC$ stop codon and $ccoP$ start codon on StuI-HindIII digested plasmid pMTSI, to yield pYO60 (Supplementary Table 1). The 3.85 kb BamHI fragment of pYO60 carrying petABC::ccoP fusion was transferred to the plasmids pBSII and pRK415 using the same sites, yielding pYO63 and pYO76, respectively. The linker-spaced versions of $petABC::ccoP$ fusion ($petABC::L2-ccoP$, $petABC::L3-ccoP$, and $petABC::L4-ccoP$) were also constructed by exchanging the StuI-HindIII fragment of pYO63 carrying petABC-ccoP with the PCR-amplified linker added versions, yielding pYO77, pYO78, and pYO80, respectively. L2, L3, and L4 linkers were introduced via the primers F-L2, F-L3, and F-L4 (Supplementary Table 1) and contained the amino acid sequences of NH$_2$-GGSGGGGSG-COOH, NH$_2$-GGSGGGGSGGGSG-COOH, and NH$_2$-ASIAGGGRTASGP-COOH, respectively. The 3.9 kb KpnI-XbaI fragments carrying these linker added versions were cloned to pRK415, yielding pYO81, pYO82, and pYO83, respectively. All constructs were subsequently verified by DNA sequencing. As the protein yields and enzymatic activities of all constructs were similar, only pYO76, which has no linker (i.e., native-like fusion), was used for subsequent work.

**Construction of *petABC::ccoP::cycY-Flag* (cyt *bc₁*-CcoP-cyt *c_y*-Flag).** Using the primers Fw-cp-cy (BalI) and Rv-cp-cy (BglII + HinIII), the C-terminally Flag-tagged *cycY* lacking its N-terminal TMH anchor (i.e., amino acid residues 1–30) was PCR amplified. The 619-bp-long PCR fragment containing *cycY-Flag* without its anchor was digested with BalI-HinIII and exchanged with its counterpart that encompasses the 3′ end of *ccoP* on pYO63 to yield the *petABC::ccoP::cycY-Flag* fusion carried by pYO91. The 4.5 kb KpnI-XbaI fragment of pYO91 containing this fusion was cloned into pRK415 using the same sites to yield pYO92 (Supplementary Table 1).

**Inactivation of CIII₂ (cyt *bc₁*) and CIV (*cbb₃*-type Cox) structural genes.** The Δ (*petABC::gm*) deletion–insertion allele carried by pYO34[32] was transferred using the gene transfer agent into the chromosome of *R. capsulatus* strain MG1 (ccoP:: kan) to yield YO12, a mutant background lacking both CIII₂ and the CcoP subunit of CIV. A similar strain, M7G-CBC1 that lacks both CIII₂ and CcoP but over-producing CcoN and CcoO of CIV[31], was also used.

**Construction of strains producing bipartite and tripartite fusion SCs.** The plasmids pYO76 and pYO92 encoding the bipartite and tripartite SCs, respectively, were conjugated into YO12 and M7G-CBC1 using triparental mating to yield pYO76/YO12 and pYO92/M7G-CBC1 strains as in ref. [65] for protein purification.

**Purification of fusion SCs.** *R. capsulatus* cells (~25–30 g from 8 L media) were resuspended in a final volume of 35–40 mL of buffer A (50 mM Tris-HCl, pH 8.0, 100 mM NaCl, 20 mM EDTA, 1 mM aminocaproic acid, and 1 mM 4-(2-aminoethyl) benzenesulfonyl fluoride hydrochloride (AEBSF) supplemented with Pierce Protease Inhibitor (Thermo Scientific, 1 mini tablet per 30 mL). Cells were disrupted by three passages through a French Pressure Cell (SLM Aminco) at 13,000 psi in the presence of 5–10 mg of DNase I (GoldBio), and cell debris was removed by centrifugation at $27,000 \times g$ for 30 min[31]. Chromatophore membranes were sedimented by centrifugation at $190,000 \times g$ for 2 h and resuspended in buffer A. The protein concentration was determined using the Pierce Protein BCA Assay Kit (Thermo Scientific), the suspension was supplemented with 10% glycerol and adjusted to a protein concentration of 10 mg/mL with buffer A. Membrane proteins were solubilized with 2% n-dodecyl-β-D-maltoside (DDM; Anatrace) for 15 min on ice, and non-solubilized materials were sedimented by centrifugation at $100,000 \times g$ for 15 min. The supernatant was subjected to chromatography, as appropriate.

The His-tagged bipartite SC was purified by anion exchange chromatography on Bio-Gel A-50 (Bio-Rad), followed by affinity chromatography on Ni-Sepharose High Performance (GE Healthcare). Solubilized proteins were loaded onto a 40 mL Bio-Gel column equilibrated with Tris buffer (50 mM Tris-HCl, pH 8.0, 100 mM NaCl, 1% glycerol, and 0.01% DDM) and washed with three column volumes (CVs) of the buffer. After another washing step with 3 CVs of Tris buffer containing 150 mM NaCl to eliminate weakly bound proteins, elution was carried out with a linear, five CV gradient from 150 to 400 mM NaCl in Tris buffer. Fractions containing both cyt *c* reductase and cyt *c* oxidase activities were combined and loaded onto a 7-mL Ni-Sepharose column equilibrated with the binding buffer (50 mM Tris-HCl, pH 8.0, 500 mM NaCl, 20 mM imidazole, 1% glycerol, and 0.01% DDM). After washing with 30 mL of binding buffer, the column was eluted with a linear 100 mL gradient from 20 to 260 mM imidazole.

The Flag-tagged tripartite SC was purified by affinity chromatography using the Anti-Flag affinity gel (Bimake). The supernatant containing solubilized proteins was loaded onto a 1 mL column equilibrated with TBS buffer (50 mM Tris-HCl, pH 7.4, 150 mM NaCl, 0.01% DDM, and 1 mM AEBSF), washed with 10 ml TBS, and eluted with 5 mL of Flag (DYKDDDDK) peptide (100 μg/mL) (Sigma) in TBS buffer.

Eluted proteins were concentrated (Amicon Ultra-15, 30 kDa molecular weight cut-off (MWCO)) (Millipore) to a final volume of 150–200 μL and loaded onto a Superose 6 Increase 10/300 GL (GE Healthcare) size exclusion column equilibrated with TBS buffer and eluted with the same buffer at a flow rate of 0.4 mL/min. Peak fractions were combined and concentrated, and PD MiniTrap G-25 columns (GE Healthcare) were used for buffer exchanges, as needed.

**Biochemical characterization of fusion SCs.** Native polyacrylamide gel electrophoresis (PAGE) was performed using 4–13% gels[66], and sodium dodecyl sulfate (SDS)-PAGE used 12.5 or 18% gels[67]. Gels were stained with Coomassie Brilliant Blue (Bio-Rad) or colloidal silver[68]. Immunoblot analyses were performed using polyclonal antibodies specific of *R. capsulatus* cyt *b*[28] and monoclonal anti-rabbit-IgG-alkaline phosphatase (Sigma) used at 1:10,000 dilution. CIV in-gel activity was revealed by incubating native gels with 0.5 mg/mL 3,3′-diaminobenzidine in 50 mM Na-phosphate, pH 7.2[69]. The *c*-type cyts were revealed by their heme peroxidase activity, using 3,3′,5,5′-tetramethylbenzidine (TMBZ) and hydrogen peroxide (H₂O₂)[70]. Briefly, gels were washed with 0.25 M Na-acetate buffer (pH 5) and incubated with 6.3 mM TMBZ in 30% methanol and 0.25 M Na-acetate, pH 5.0. After 1-h incubation, 0.4% H₂O₂ were added to reveal the peroxidase activity of *c*-type cyts.

Reduced *minus* oxidized spectra were recorded in 50 mM MOPS, pH 7.0 using a Cary 60 UV-Vis spectrophotometer (Agilent). Usually, 50 μg of purified protein

was fully oxidized by adding a grain of K₃[Fe(CN)₆] and then gradually reduced with a few grains of sodium ascorbate or sodium dithionite, as appropriate. Protein concentrations were determined with a NanoDrop 2000c (Thermo Scientific) using the A₂₈₀ method[71].

Cyt *c* reductase (cyt *bc₁*) activity was determined as described[72]. Accordingly, 10 mM stock solution of 2,3-dimethoxy-5-methyl-6-decyl-1,4-benzoquinone (DB) (Sigma) in dimethyl sulfoxide was reduced to DBH₂ with a few grains of sodium borohydride and excess borohydride was quenched by adding HCl to a final pH of 6.0. For assays, 40 μM DBH₂ were added to 25 μM horse heart cyt *c* (Sigma) in 500 μL assay buffer (40 mM sodium phosphate, pH 7.4, 20 mM sodium malonate, 0.5 mM EDTA, 0.5 mM KCN, and 0.01% DDM) in a stirred cuvette, and the non-enzymatic rate of cyt *c* reduction was recorded at 550 nm for 1 min (Cary 60 UV-Vis spectrophotometer, Agilent). The reaction was initiated by adding 1–2 μg of purified protein and cyt *c* reduction was monitored. The enzymatic activity was inhibited by adding 1 μM stigmatellin (Fluka), inhibitor of cyt *bc₁*-type CIII₂.

The CIV (cyt *cbb₃*-Cox) activity was determined as described[73]. Reduced horse heart cyt *c* (Sigma) was prepared by incubating a 1–2 mM stock solution with 10 mM sodium dithionite for 15 min at room temperature, excess dithionite was removed by gel filtration using a PD-10 desalting column (GE Healthcare), and the final concentration of reduced cyt *c* was calculated based on its absorption at 550 nm and an $\varepsilon = 20 \text{ mM}^{-1} \text{ cm}^{-1}$. In all, 20 μM reduced cyt *c* in 500 μL assay buffer (10 mM Tris-HCl, pH 7.0, 100 mM KCl) were prepared in a stirred cuvette (Cary 60 UV-Vis spectrophotometer), and the baseline was recorded for 1 min. The reaction was started by adding 1–2 μg of purified protein, and cyt *c* oxidation monitored at 550 nm. The enzymatic activity was inhibited by addition of 250 μM KCN as an inhibitor of *cbb₃*-Cox.

DBH₂-dependent oxygen consumption was monitored using a mini Clark-type oxygen electrode (Instech Laboratories, PA) at 25 °C. The assay was performed in the same buffer as that used for cyt *c* reductase activity (40 mM sodium phosphate, pH 7.4, 20 mM sodium malonate, 0.5 mM EDTA, and 0.01% DDM) but without KCN. A baseline was recorded after adding 100 μM DBH₂ to 1 mL of assay buffer, and the reaction was started by adding 30 μg of purified protein. As needed, purified protein was pre-incubated with a 1:1 (w/w) mixture of *E. coli* polar lipids (Avanti), which improved the activity. In all, 20 μM horse heart cyt *c* (Sigma), 250 μM KCN, or 1 μM of stigmatellin were used as specificity controls. For all measurements, three biological and three technical repeats are used.

**Identification of protein subunits by MS.** Protein subunits were identified by MS using appropriate gel bands reduced (dithiothreitol (DTT); Sigma), alkylated (iodoacetamide; Bio-Rad) and subjected to in-gel trypsin digestion (Promega, Sequencing Grade Modified Trypsin) overnight at 37 °C. Peptides eluted from the gel samples were dried, desalted using ZipTips (Millipore U-C18 P10, Millipore), lyophilized, and stored at −80 °C. They were resuspended in 10 μL 5% acetonitrile (ACN)/0.1% formic acid prior to MS, using either a LCQ Deca XP + ion trap, or a Q-Exactive Quadrupole-Orbitrap mass spectrometer (both from Thermo Fisher Scientific)[74]. The LCQ Deca XP+ mass spectrometer was coupled to a Thermo-Dionex LC Packings Ultimate Nano HPLC system controlled by the Thermo Xcalibur version 2.0 software. Peptides were separated on a 15 cm C18 nano-column (Thermo-Dionex, NAN-75-15-03-PM) using a 45-min linear gradient from 4 to 40% buffer B (100% ACN, 0.1% formic acid), followed by a 7-min gradient from 40 to 80% buffer B and 8-min wash with 80% B (constant flow rate 150 nL/min). MS/MS data were acquired in data-dependent analysis mode with dynamic exclusion enabled (repeat count: 3, exclusion duration: 3 min). Full MS survey scans (mass range 300–2000 *m/z*) were followed by MS/MS fragmentation (normalized collision energy 35) of the top 3 most intense ions. The Q-Exactive Quadrupole-Orbitrap mass spectrometer was coupled to an Easy-nLC™1000 nano HPLC (Thermo Fisher Scientific), and samples were loaded in buffer A (0.1% formic acid) onto a 20-cm-long fused silica capillary column (75 μm ID), packed with reversed-phase Repro-Sil Pur C18-AQ 3 μm resin (Dr. Maisch GmbH, Ammerbuch, Germany). Peptides were eluted using a 45-min linear gradient from 4 to 40% buffer B (100% ACN, 0.1% formic acid), followed by a 7-min gradient from 40 to 80% buffer B and 8-min wash with 80% B (constant flow rate 300 nL min). The Q-Exactive was operated in data-dependent acquisition mode with dynamic exclusion enabled (repeat count: 1, exclusion duration: 20 s). Full MS survey scans (mass range 300–1600 *m/z*) at high resolution (70 000 at 200 *m/z*) were followed by MS/MS fragmentation of the top 15 most intense ions with higher energy collisional dissociation at a normalized collision energy of 22 (resolution 17 500 at 200 *m/z*). Dual lock mass calibration was enabled with 371.101233 and 445.120024 *m/z* background ions.

MS spectra were searched against the *R. capsulatus* protein database (https://www.uniprot.org, last modified 01/15/2020; *R. capsulatus* (strain ATCC BAA-309/NBRC 16581/SB1003)) using Proteome Discoverer 1.4 (Thermo Fisher Scientific) with Sequest-HT search engine. Search parameters were set to full trypsin digestion, with maxima of three missed cleavages and three modifications per peptide. Oxidation of methionine (+16 Da) and carbamidomethylation of cysteine (+57 Da) were selected as dynamic modifications. Precursor and fragment ion tolerances were set to 2 and 1 Da, respectively, for the LCQ DecaXP+, and 10 ppm and 0.6 Da, respectively, for the Q-Exactive data. False discovery rates (FDRs) by target-decoy search were set to 0.01 and $X_{corr}$ filter based on charge (*z*) were: >2 for

$z = 2$, >2.5 for $z = 3$, and >2.6 for $z = 4$[74]. All identifications are listed in Supplementary Table 2.

**Binding of cyt $c_y$ and soluble cyt S-$c_y$ to purified proteins.** Binding of purified cyt $c_y$[75] or a variant of it containing only the soluble cyt domain (cyt S-$c_y$) (residues 99–199) to the bipartite SC was assayed by mixing a 5–10-fold molar excess of purified cyt $c$ with 125 µg of purified SC in a final volume of 150 µL TBS buffer (50 mM Tris-HCl, pH 7.4, 150 mM NaCl, and 0.01% DDM). The mixture was separated by chromatography using Superose 6 Increase size exclusion column, and elution fractions were analyzed by SDS-PAGE followed by silver staining. Cyt S-$c_y$ was purified from the *R. capsulatus* strain pYO135/FJ2-R4 (Supplementary Table 1) grown under photosynthetic conditions to maximize its yield[32]. Cells were washed with 20 mM Tris-HCl, pH 8.0, and resuspended 1:5 (w/v) in the same buffer supplemented with 50 mM NaCl. Polymyxin B sulfate (1 mg/mL) was added, and the cell suspension was incubated for 75 min on ice with gentle stirring. After centrifugation for 20 min at 10,000 × g followed by 3 h at 150,000 × g, the supernatant was collected and concentrated to a final volume of 2 mL (Amicon Ultra-15 3 kDa MWCO) (Millipore). Aliquots of 300 µL were loaded onto Superose 6 Increase (GE Healthcare) equilibrated in 30 mM Tris-HCl, pH 8.0, and eluted in the same buffer. Fractions eluting after 20 mL were combined and concentrated using Amicon Ultra-15 3 kDa MWCO filters (Millipore).

**Binding of cyt $c_2$ to purified proteins.** Binding of cyt $c_2$ to $cbb_3$-type CIV was determined by mixing 2-fold molar excess of purified *R. capsulatus* cyt $c_2$[76] with 300 µg of purified CIV[31] under low-salt conditions (20 mM Tris-HCl, pH 7.4, 1 mM NaCl, and 0.01% DDM). The mixture (250 µL total volume) was loaded onto a Superose 6 Increase (GE Healthcare) sizing column equilibrated with the same buffer to separate the proteins. Elution fractions were concentrated (Amicon Ultra-15, 3 kDa MWCO) (Millipore), and a volume containing 1–5 µg of protein was analyzed by SDS-PAGE followed by silver staining.

**Protein crosslinking using chemical crosslinkers.** For crosslinking, 180 µg of purified bipartite SC at a concentration of 1 mg/mL in phosphate-buffered saline buffer (50 mM Na-Phosphate, pH 7.4, 150 mM NaCl, and 0.01% DDM) mixed with 5-fold molar excess (90 µg) of purified cyt $c_y$[75] were supplemented with 6 mM DSBU (Thermo Fisher Scientific) and incubated on ice for 2 h. The reaction was quenched by adding 50 mM of ammonium bicarbonate, and the mixture was analyzed by MS. In all, 300 µg of purified $cbb_3$-type CIV[31] were mixed with a 2-fold molar excess of purified cyt $c_2$[76] at a final protein concentration of 200 µg/mL under low-salt conditions (20 mM Na-phosphate, pH 7.4, 1 mM NaCl, and 0.01% DDM), and incubated with 20 mM 4-(4,6-Dimethoxy-1,3,5-triazin-2-yl)-4-methylmorpholinium chloride (DMTMM) (Sigma) for 1 h at room temperature. Similarly, 300 µg of purified $bc_1$-type CIII$_2$ were mixed with 2-fold molar excess of purified cyt $c_2$[76] and treated with DMTMM as above. In both cases, the reaction was stopped by removing excess DMTMM with a PD MiniTrap G-25 desalting column (GE Healthcare), and crosslinked proteins were precipitated with 20% (w/v) tri-chloroacetic acid (TCA, Sigma) at 4 °C for 1 h. Proteins were pelleted by centrifugation at 21,000 × g for 15 min and washed with 10% TCA in 0.1 M Tris-HCl and then with acetone (Fisher). The solvent was discarded, and the pellet was air-dried and then stored at −80 °C for analysis by MS.

**MS of crosslinked proteins.** Crosslinked proteins were resuspended in an appropriate volume of solution A (2.5% SDS and 50 mM triethylammonium bicarbonate final concentrations) and reduced with 10 mM DTT (US Biological) for 30 min at 30 °C, followed by alkylation with 50 mM iodoacetamide (Sigma Aldrich) for 30 min at 30 °C. The proteins were processed using an S-Trap™ according to the protocol recommended by the supplier (Protifi, C02-mini) and digested with trypsin (Thermo Fisher Scientific) in 1:10 (w/w) enzyme/protein ratio for 1 h at 47 °C. Peptides eluted from this column were vacuum-dried and resuspended with the peptide fractionation–elution buffer for liquid chromatography-MS (LC-MS; (70% (v/v) LC-MS-grade water (Thermo Fisher Scientific), 30% (v/v) ACN (Thermo Fisher Scientific) and 0.1 % (v/v) trifluoroacetic acid (TFA, Thermo Fisher Scientific)]. Peptides were fractionated using AKTA Pure 25 with Superdex 30 Increase 3.2/300 (GE Life Science) at a flow rate of 30 µL/min of the elution buffer, and 100 µL fractions were collected. Based on the elution profile, fractions containing enriched crosslinked peptides of higher molecular masses were vacuum-dried and resuspended with LC-MS-grade water containing 0.1% (v/v) TFA for MS analysis. One half of each fraction was analyzed by a Q-Exactive HF mass spectrometer (Thermo Fisher Scientific) coupled to a Dionex Ultimate 3000 UHPLC system (Thermo Fischer Scientific) equipped with an in-house made 15 cm long fused silica capillary column (75 µm ID), packed with reversed-phase Repro-Sil Pur C18-AQ 2.4 µm resin (Dr. Maisch GmbH, Ammerbuch, Germany). Elution was performed using a gradient from 5 to 45% B (90 min), followed by 90% B (5 min), and re-equilibration from 90 to 5% B (5 min) with a flow rate of 400 nL/min (mobile phase A: water with 0.1% formic acid; mobile phase B: 80% ACN with 0.1% formic acid). Data were acquired in data-dependent MS/MS mode. Full scan MS settings were: mass range 300–1800 $m/z$, resolution 120,000; MS1 AGC target 1E6; MS1 Maximum IT 200. MS/MS settings were: resolution 30,000; AGC target 2E5; MS2 Maximum IT 300 ms; fragmentation was enforced by higher-energy collisional dissociation with stepped collision energy of 25,

27, 30; loop count top 12; isolation window 1.5; fixed first mass 130; MS2 Minimum AGC target 800; charge exclusion: unassigned, 1, 2, 3, 8, and >8; peptide match off; exclude isotope on; dynamic exclusion 45 s[43]. Raw files were converted to mgf format with TurboRawToMGF 2.0.8[77].

MeroX 2.0.1.4[44,46] (for MS-cleavable and non-cleavable crosslinkers, http://www.stavrox.com/) was used as the main search engine for both DMTMM and DSBU. Additionally, FindXL[46,47] (for non-cleavable crosslinkers, http://biolchem.huji.ac.il/nirka/software.html) was used to analyze DMTMM spectra, and MassAI 19.07[78] (for MS-cleavable and non-cleavable crosslinkers, http://www.massai.dk) was used to analyze DSBU spectra to cross-validate results from MeroX in each case. Only the crosslinked peptides that were cross-validated by two different search engines (i.e., MeroX and FindXL for DMTMM and MeroX and MassAI for DSBU) were used for docking experiments (Supplementary Table 4). MeroX was run in RISEUP mode, with default crosslinker mass and fragmentation parameters for DSBU and in Quadratic mode with default crosslinker mass parameters for DMTMM; precursor mass range, 300–10,000 Da; minimum precursor charge 4; precursor and fragment ion precisions 5.0 and 10.0 ppm, respectively; maximum number of missed cleavages 3; carbamidomethylation of cysteine and oxidation of methionine, as fixed and variable modifications, respectively; results were filtered for score (>10) as well as the MeroX built-in CL spectral matches FDR (<1%). The database consisted of the proteins listed in Supplementary Table 2, supplemented with the assembly factors such as CcoH and the 116 proteins of the cRAP database (www.thegpm.org; contaminant proteins commonly occurring in MS analyses). The latter increased the size of the decoy database to yield more reliable FDRs. FindXL was used to analyze and validate MeroX results for DMTMM CLs. The default FindXL parameters were used as described before[47] with the possible CL amino acids adjustments for K, Y, S, or T on one peptide and E or D on the other peptide. MassAI was used to validate CLs identified by MeroX for DSBU with standard settings, except 5 ppm MS accuracy, 0.05 Da MS/MS accuracy, 3 allowed missed cleavages, and carbamidomethylation of cysteine and oxidation of methionine, as fixed and variable modifications, respectively. Additional filters were: # fragments A ≥4, # fragments B ≥4, Intensity ≥1000, Score ≥10. Databases used in FindXL and MassAI consisted of the proteins listed in Supplementary Table 2, supplemented with the assembly factors, such as CcoH. Visualization of the CLs in the structures used Chimera[79] with the Xlink Analyzer plug-in[80].

**Negative staining and cryo-EM sample preparation.** For negative staining, purified SCs were diluted to 0.01–0.05 mg/mL concentrations in TBS (50 mM Tris-HCl, pH 7.4, 150 mM NaCl, 0.01% DDM, and 1 mM AEBSF) buffer, 5 µL applied to glow-discharged (20 s, 25 mA, Pelco easiGlow) carbon-coated Cu grids (CF300-CU, EMS), incubated for 1 min, and stained with 2% uranyl acetate. Grids were imaged on FEI Tecnai 12 transmission electron microscope operating at 120 kV, with a CCD camera (Gatan BM-Ultrascan).

For cryo-EM, the tripartite SC was used as purified, whereas the purified bipartite SC was mixed with a 5-fold molar excess of purified cyt $c_y$[75]. In all, 2.5 µL of the mixture, containing 3–5 mg/mL proteins, were applied to CFlat holey carbon grids (1.2/1.3–400 mesh or 2/2–300 mesh) (EMS), which were glow discharged (2 min, 25 mA, Pelco easiGlow) before sample application. Grids were blotted for 9 s at force 0 (CFlat-1.2/1.3) or for 3 s at force −5 (CFlat-2/2) and flash-frozen in liquid ethane cooled with liquid nitrogen using FEI Vitrobot Mark IV (25 °C, 100% humidity). Plunge freezing conditions were optimized using FEI Tecnai TF20 TEM operating at 200 kV, equipped with an FEI Falcon II camera.

**Cryo-EM data acquisition and processing.** All cryo-EM grids were imaged using FEI Titan Krios electron microscope operating at 300 kV. Images of the tripartite SC (SC~$c_y$) were recorded using a Gatan K2 Summit direct electron detector, equipped with an energy quantum filter (20 eV), and operated in super-resolution mode at a nominal magnification of ×105,000, resulting in a binned pixel size of 1.32 Å. Images were dose-fractionated to 40 frames with a total exposure time of 10 s, and a total dose of 40 e−/A$^2$. Automated data acquisition was carried out using the Latitude software (Gatan), and nominal underfocus values varied from 1 to 2.5 µm. Movies were motion corrected using MotionCor2[81] and CTF parameters were determined with CTFFIND 4.1[82]. About 10,000 particles were manually picked and subjected to an initial reference-free two-dimensional (2D) classification using Relion 3.0[83]. Representative classes were selected and used as a template for auto-picking. After sorting and two rounds of 2D classification, ~30,000 particles were retained from the dataset. For 3D classification, an initial model was created using *R. capsulatus* CIII$_2$ structure (PDB: 1ZRT) and low-pass filtered to 60 Å using EMAN2[84]. A 3D map with a nominal resolution of ~11 Å containing a total of ~12,000 particles (40%) was obtained, showing a CIII$_2$ associated on one side with a single copy of CIV. For further analyses, additional datasets were collected using samples from the same batch and identical imaging conditions. From a total of ~17,000 images, ~1,000,000 particles were automatically picked using the template obtained from the first dataset, and after sorting and two rounds of 2D classification, ~500,000 particles were retained. Various subsets of all images were processed separately and yielded the same overall results with slightly different versions of individual maps. For clarity, only the paths to the best representative of each of the final maps are shown (Supplementary Fig. 2a–c). For 3D classification, the map obtained from the first dataset (Supplementary Fig. 2, Box 3) was low-pass filtered to 60 Å and used as the initial model. Of the five classes

obtained, one showed clear features of a $CIII_2$ dimer associated with a single monomer of CIV (Supplementary Fig. 2, class005 in a, class002 in b and c). The remaining classes showed the same overall shape but lacked density or resolution in different parts of the structure. Only two of the five maps contained a second monomer of CIV associated with the other monomer of $CIII_2$ (Supplementary Fig. 2c). These two classes showing this feature were combined, yielding a subset of ~220,000 particles. These particles were further processed by another round of 2D classification and subjected to 3D classification, yielding three classes (Supplementary Fig. 2c). Only one of the maps obtained in this second round of 3D classification clearly showed density for a second monomer of CIV. Aligned particles from this group were subclassified into six classes in another round of 3D classification using a soft mask and no image alignment. The low number of particles (~5,000) per class limited the resolution of the maps (<10 Å) and could not be further improved through 3D refinement.

Refinement of the individual maps (Supplementary Fig. 2a, b) led to relatively low-resolution reconstructions, particularly in the CIV portion of the map, probably due to greater structural heterogeneity in the sample with respect to this portion of the tripartite SC. The highest resolution was obtained with a map containing a single copy of CIV by combining all classes showing well-defined features corresponding to it and subjecting them to a second round of 3D classification, focused on CIV portion of the map. For this purpose, a soft mask around CIV was created by fitting the closely related structure of *P. stutzeri* (PDB: 3MK7) into the map and low-pass filtering it to 10 Å. All information outside of this mask was subtracted from the aligned particles, and the remaining particles were subjected to a masked classification without image alignment, to yield six classes (Supplementary Fig. 2a). Alternatively, a cylindrical mask was wrapped around the CIV portion of the map and used in a similar procedure (Supplementary Fig. 2b). In each case, classes showing the highest level of details were retained, and the entire unsubtracted particle dataset was subjected to 3D auto-refinement followed by per-particle CTF refinement, Bayesian polishing, and post-processing using Relion 3.0. The two extreme conformations of the orientation of CIV relative to $CIII_2$ were represented by the maps SC-1A and SC-1B (Supplementary Fig. 2a, b), which were refined to 6.1 and 7.2 Å resolutions, respectively (Supplementary Table 3).

Images of the bipartite SC supplemented with cyt $c_y$ (SC + $c_y$) were recorded by a Gatan K3 direct electron detector equipped with an energy quantum filter (20 eV) and operated in counting mode at a nominal magnification of ×64,000, corresponding to a pixel size of 1.36 Å. Images were dose-fractionated to 80 frames with a total exposure time of 3.1 s and a total dose of 40 $e^-/A^2$. Nominal underfocus values varied from 1 to 2.5 μm. Automated data acquisition was carried out using the Latitude software (Gatan) and image shift (~ 2 μm, 4 images per stage position) was used for accelerated data collection. For this sample, 5480 images were collected in the first session, motion corrected using MotionCor2[81], and CTF parameters were determined with CTFFIND 4.1[82]. For auto-picking, the template previously obtained with the tripartite SC was used, with this and all subsequent steps done in Relion 3.0[83]. After sorting and two rounds of 2D classification, ~228,000 particles were retained (Supplementary Fig. 5a). The same initial model as for the tripartite SC was used for 3D classification into five classes. Two classes (001 and 002) resembled more to the overall shape of a $CIII_2$ without CIV, while the other three classes showed the same overall shape of a $CIII_2$CIV SC, as seen with the tripartite sample. In all, 37,460 particles corresponding to the SC class with the highest level of detail were combined with SC particles from the second dataset (Supplementary Fig. 5b) to yield the best final map of the bipartite SC. The second dataset consisted of 12,200 images, which were collected and processed under the same conditions as the first dataset. In all, 1.6 million particles were auto-picked, and after 2 rounds of 2D classification, classes were split into ~340,000 large particles likely representing $CIII_2$CIV and ~465,000 particles resembling $CIII_2$ (Supplementary Fig. 5b, d). The former particles were subjected to 3D classification into three classes, and the class most similar to the overall shape of $CIII_2$CIV was identified. It contained ~118.000 particles, but due to its low level of detail, it was subjected to another round of 2D classification. The 34,819 particles thus retained were combined with 37,460 SC particles from the first dataset (Supplementary Fig. 5a). After 3D classification into five classes, the class with the highest level of detail contained 56% of particles and clearly showed the overall shape of $CIII_2$CIV. In all, 41,017 aligned particles corresponding to this class were extracted and subjected to a second round of 3D classification without image alignment using a soft mask. Of the six 3D classes, the one with the highest nominal resolution contained 14,978 particles (36%) and was subjected to 3D auto-refinement and post-processing, followed by per-particle CTF refinement and Bayesian polishing. After a second round of 3D auto-refinement and post processing, the final map (SC-2A) of the bipartite $CIII_2$CIV was obtained at a nominal resolution of 5.18 Å (Supplementary Table 3). This map was very similar to that (SC-1A) of the tripartite $CIII_2$CIV, except for its higher resolution and strongly improved density and resolution of the extra TMHs at the interface. Unlike the tripartite samples, no major class corresponding to the conformation seen in SC-1B was identified.

As the bipartite SC samples contained a significant amount of $CIII_2$ particles without CIV, subsets of smaller 2D classes consistent with a $CIII_2$ dimer were selected and processed separately. Of the ~376,000 total particles retained from the dataset 1, ~267,000 were identified as resembling $CIII_2$. After a second round of 2D classification (Supplementary Fig. 5c), ~213,000 particles were retained and used to generate an ab initio 3D model. This model was low-pass filtered to 60 Å and used

as initial model for the 3D classification of ~465,000 $CIII_2$ particles from dataset 2 (Supplementary Fig. 5d). Particles corresponding to the class with the highest resolution and level of detail (~185,000) were combined with the $CIII_2$ particles from dataset 1 to yield ~400,000 particles and subjected to 3D classification. The ~170,000 particles corresponding to the best class were further processed following two different strategies. $CIII_2$ being a homodimer based on X-ray structure, C2 symmetry was applied in the next round of 3D classification and in all subsequent steps (Supplementary Fig. 5e). The best class contained ~120,000 particles (70%), which were extracted and, similar to $CIII_2$CIV, subjected to another round of 3D classification without image alignment using a soft mask. Of the six classes obtained, some showed the ED of both monomers of the FeS protein close to heme $b_L$ ("**b** position", b-b) while other classes showed both EDs close to heme $c_1$ ("**c** position", c-c). Individual classes were subjected to 3D auto-refinement and post-processing, followed by two rounds of per-particle CTF refinement, Bayesian polishing, 3D auto-refinement, and post-processing. Only the final map with the highest nominal resolution (3.30 Å) is shown (Supplementary Fig. 5e, $CIII_2$; Supplementary Table 3). This map contained 37,997 particles and showed both FeS-EDs close to **b** position but at a lower local resolution and occupancy than the rest of the map, indicating structural flexibility and conformational heterogeneity. A similar approach was used in Supplementary Fig. 5f, except that no C2 symmetry was applied in the two rounds of 3D classifications. Of the three best classes obtained in the second round, one showed both EDs in the **b** position, one showed both in the **c** position, and one showed a heterodimeric conformation with one monomer in the **b** and the other in the **c** position. Each of the maps was further processed as in Supplementary Fig. 5e, but C2 symmetry was only applied in case of the homodimeric structures. The final maps were $CIII_2$ c-c with both EDs in the **c** position at 3.8 Å, $CIII_2$ b-b with both ED's in the **b** position at 3.5 Å, and $CIII_2$ b-c with one ED in the **b** and the second in the **c** position at 4.2 Å (Supplementary Table 3). The nominal resolutions thus obtained were slightly lower than in map $CIII_2$ (Supplementary Fig. 5e). However, by omitting C2 symmetry application to the 3D classifications, the conformational heterogeneity of the FeS protein EDs was resolved, and a subset of particles showing a heterodimeric conformation was identified. The different locations (i.e., **b** or **c** positions) of the EDs were clear in the three maps (Supplementary Fig. 5f), but their occupancy and local resolutions was lower than the rest of the structure.

A third dataset was also collected and yielded ~303,000 particles after the first 2D classification. Adding these particles to the first two datasets did not improve the maps shown in Supplementary Fig. 5a, b, but it turned out to be informative. As shown in Supplementary Fig. 5g, particles from dataset 3 were subjected to two rounds of 3D classification. Interestingly, an extra density near the periplasmic domain of CcoP could be seen in some classes. This was not observed in any subclass obtained from the datasets 1 and 2 and tentatively thought to correspond to the cyt domain of cyt $c_y$. Focused classification followed by 3D auto-refinement and post-processing, using wider soft masks around the periplasmic domain of CcoP to avoid cutting off any of the extra density, led to a final map (SC-2B) with limited resolution (10.5 Å) and could not be further improved by sub-classification due to the low number of particles in each subclass.

**Refinement of *R. capsulatus* $CIII_2$ in the cryo-EM maps**. The X-ray based structure of *R. capsulatus* $CIII_2$ (PDB: 1ZRT) was fitted into map $CIII_2$ (EMD-22189), which had the highest nominal resolution of all maps obtained (Supplementary Fig. 5e), and refined using Phenix1.16[85]. The real space refinement approach included four rounds of global minimization, local grid search, morphing, and simulated annealing, with the final round also including ADP (B-factor) refinement. Each round included five cycles using default settings, and morphing and annealing were performed in each cycle. Secondary structures were determined by Phenix1.16, using default search settings, and restrictions were applied during real space refinement. Due to the low occupancy and limited local resolution, the FeS-ED proteins (residues 50–191) were only subjected to rigid body fitting followed by two rounds of global minimization and local grid search (5 cycles each) but not to morphing and simulated annealing. To ensure the correct cofactor geometry, hemes and [2Fe-2S] clusters including their coordinating residues were copied from the high-resolution X-ray structure of the homologous $CIII_2$ (cyt $bc_1$) from *R. sphaeroides* (PDB: 6NHH). Validation was performed using MolProbity[86] (http://molprobity.biochem.duke.edu/), and outliers (Ramachandran, rotamer, bonds, angles) were manually corrected in Coot[87], using real space refinement and regularization. The model that was refined in map $CIII_2$ (EMD-22189) was subsequently used for rigid body fitting into the maps $CIII_2$ b-b, $CIII_2$ c-c, and $CIII_2$ b-c (Supplementary Fig. 5f and Supplementary Table 3). Each chain was treated as one separate body, except the FeS protein, which was split into its TMH (11–49) and ED (50–191) residues. After the procedure, the linker region between the ED and TMH (residues 40–50) was remodeled in Coot using real space refinement and regularization.

**Modeling of *R. capsulatus* $cbb_3$-type CIV subunits**. *R. capsulatus* $cbb_3$-type CIV was modeled using *P. stutzeri* $cbb_3$-type CIV structure (PDB: 5DJQ or 3MK7) (sequence identities for *R. capsulatus* CcoN: 68%, CcoO: 55%, CcoP: 34%) as a template, and comparative models were computed using MODELLER v9.18[88] with defined secondary structure and crosslinking restraints. The secondary structures for the regions without any template coverage (CcoP residues 104–112, 176–178, 184–197, and 204–215; CcoO residues 172–185 and 192–199), due to the insertion

sequences that are present only in *R. capsulatus* CIV, were estimated using PsiPred Protein Sequence Analysis Workbench[89,90], and added as restraints to MODELLER. Cross-link distances were added as Gaussian restraints to MODELLER with a mean of 18.0 Å and a standard deviation of 1.0 Å. Problematic loop regions (a total of ten loops longer than four amino acids) were detected by MolProbity[86,91] and remodeled using MODELLER "slow" refinement method (Supplementary Table 3). As *R. capsulatus* CcoN is longer than that of *P. stutzeri* with a predicted extra N-ter TMH, whereas CcoP is shorter with only one predicted N-ter TMH instead of two, these regions were not included into the model. The regions CcoO 179–214 as well as CcoP 1–12, 53–59, 161–173, and 272–280 without template coverage that were not supported by the cryo-EM maps were omitted.

**Modeling of *R. capsulatus* CcoH and cyt $c_y$.** For CcoH (residues Met 1 to Thr35), five ab initio models were obtained using I-TASSER server[92]. All models were almost identical, and one with the lowest energy score was retained. For *R. capsulatus* cyt $c_y$, its homologs with known structures were detected using HHpred[93], and the structure of cyt domain of cyt $c_{552}$ from *P. denitrificans* (PDB: 3M97; sequence identity: 61%) was used as a template for the soluble cyt domain of $c_y$, and comparative models were computed using MODELLER[88].

**Modeling of the extra N-ter TMH of CcoN (TMH0).** A 29-residue model consisting of CcoN residues Arg25 to Asp53, which includes the predicted transmembrane region (Met30-Leu48, UniProtKB, D5ARP4), was obtained from the I-TASSER server[94]. The alpha helical region included residues Leu27 to Thr50, and the model including the residues Arg25 to Leu48 was manually fitted into the density map for visualization purpose in Fig. 2. Related coordinates were not deposited in the PDB because the registration could not be determined due to the lack of side chain density.

**Modeling of cyt $c_y$ and CcoH TMH interactions.** The 30-residue-long cyt $c_y$ TMH was manually docked into the density map by positioning Phe15 and Tyr21 into the corresponding densities and refined by real-space refinement and regularization using Coot. The I-TASSER model of CcoH was docked into the map by moving it along the corresponding density, retaining its close association with cyt $c_y$ TMH. The best fit was found when Ala23 and Val24 of CcoH were located at the interface with cyt $c_y$ TMH. The interaction of the two helices was optimized by GalaxyWEB[95], and the model with the lowest energy profile was refined in map SC-2A using Phenix 1.17. The refinement strategy included five cycles of global minimization, rigid body fitting (where cyt $c_y$ and CcoH chains were treated as separate bodies), local grid search, and ADP (*B*-factor) refinement with default settings for all steps. Secondary structure restrictions were applied to the α-helical parts of CcoH as predicted by I-TASSER (residues 12–34) and to cyt $c_y$ as predicted by GalaxyWEB (residues 2–9 and 12–29). CIII$_2$ and CIV models were present during the refinement to keep the rest of the map occupied, but changes made during this procedure were discarded as these models were refined separately.

**Integrative modeling of SC.** The entire SC was assembled by an integrative modeling using the cryo-EM map, XL-MS, co-evolutionary analysis, and subunit models described above. CIII$_2$ and CIV models were fitted into the cryo-EM maps in UCSF Chimera[79].

**Docking of CcoH, cyt $c_y$, and cyt $c_2$ to bipartite SC.** Additional data about the interaction interfaces between the different subunits of the SC were obtained using RaptorX-ComplexContact[33] for each pair of the SC subunits. Based on co-evolution and machine learning, this method predicted the pairs of residues that are in contact. The PatchDock, which is an efficient rigid docking method that maximizes geometric shape complementarity[34], was used to generate docked configurations of CcoH, cyt $c_2$, and cyt $c_y$ to CIII$_2$ and CIV, as appropriate, and in all docking analyses the top 50 models were retained. The different subunits were docked in parallel independently from each other.

**Docking of CcoH to CIV.** A total of 16 contacts predicted by RaptorX-ComplexContact, with probabilities >0.5 were used as distance restraints for protein–protein docking[34]. The models satisfied 13 of the contacts, and a single cluster of docked models evidenced by the convergence of the top 50 results was obtained (Supplementary Fig. 7). This cluster coincided with the additional feature seen at the SC interface, and CcoH TMH was modeled into this feature.

**Docking of cyt $c_2$ to CIII$_2$ and CIV.** *R. capsulatus* cyt $c_2$ of known structure (PDB: 1C2N) was docked to CIII$_2$ using the three distance restraints between cyt $c_2$ and cyt $c_1$ derived from the protein crosslinking data (Supplementary Table 4). Similarly, cyt $c_2$ was docked to CIV using nine distance restraints provided by XLs (1 CL to CcoP and 8 to CcoO, Supplementary Table 4). The models yielded one main cluster in each case and covered 100 and 89% of the data for CIII$_2$ and CIV, respectively.

**Docking of cyt $c_y$ to CIII$_2$CIV SC.** The model of cyt domain of $c_y$ was docked by PatchDock to bipartite CIII$_2$CIV, which has both FeS-EDs of CIII$_2$ in the **b** position. Ten distance restraints derived from the XL-MS data (6 with DMTMM with max limit of 30 Å and 4 with DSBU with max limit of 35 Å, Supplementary Table 4) were used, top models yielded two main clusters (1 and 2) of the docked models on the *p* side of CIII$_2$ (Supplementary Fig. 4), and satisfied 100% of the restraints. As the cryo-EM data revealed that CIII$_2$ particles could have their FeS-EDs in the **c** position, the corresponding CIII$_2$ c-c, CIII$_2$ b-b, and CIII$_2$ b-c models were used for docking via PatchDock the cyt domain of $c_y$ onto CIII$_2$.

All the models were ranked using statistically optimized atomic potentials (SOAP)[96], and those that have low SOAP scores were retained.

**Reporting summary.** Further information on research design is available in the Nature Research Reporting Summary linked to this article.

## Data availability

*R. capsulatus* database used is: https://www.uniprot.org, last modified 01/15/2020; *R. capsulatus* strain ATCC BAA-309/NBRC 16581/SB1003. All structural data and corresponding cryo-EM maps that support the findings of this study have been deposited to PDB and EMDB under the accession codes: CIII$_2$ (PDB: 6XI0; EMD-22189); CIII$_2$ c-c (PDB: 6XKT; EMD-22224); CIII$_2$ b-c (PDB: 6XKU; EMD-22225); CIII$_2$ b-b (PDB: 6XKV; EMD-22226); SC-2A (PDB: 6XKW; EMD-22227); SC-1A (PDB: 6XKX; EMD-22228); SC-1B (PDB: 6XKZ; EMD-22230). XL-MS data have been deposited to ProteomeXchange PRIDE repository with the dataset identifier PXD020038. All other relevant data are available upon request from the corresponding authors. Source data are provided with this paper.

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

## Acknowledgements

This work was supported partly by the NIH grants GM 38237 to F.D., GM123233 to K.M., GM110174 and AI118891 to B.A.G., T32-GM008275 to T.V., T32-GM071339 to H.J.K., and partly by the Division of Chemical Sciences, Geosciences and Biosciences, Office of Basic Energy Sciences of Department of Energy grant DE-FG02-91ER20052 to F.D., by ISF 1466/18, BSF 2016070, and Ministry of Science and Technology 80802 grants to D.S.-D. Y. O. was supported by the grant GRK2202-23577276/RTG from DFG, Germany. Data analysis was partly supported by the NIH grant S10OD023592. This research was supported in part by the NCI, National Cryo-EM Facility at the Frederick National Laboratory for Cancer Research under contract HSSN261200800001E. The authors thank Ulrich Baxa, Thomas Edwards, and Adam Wier for their support and helpful discussions. Some cryo-EM data were also obtained at the University of Massachusetts Cryo-EM Core Facility, and we thank Dr. Chen Xu, Dr. KangKang Song, and Dr. Kyounghwan Lee for their support. Cryo-EM sample screening and optimization was performed at the Electron Microscopy Resource Laboratory at the Perelman School of Medicine, University of Pennsylvania, and we thank Dr. Sudheer Molugu for his support. We also thank Dr. S. Saif Hasan, Dr. Brian G. Pierce, and Dr. Christian Presley at the Institute for Bioscience and Biotechnology Research, University of Maryland for insightful discussions and invaluable help they provided during this study. S.S. and F.D. thank Vivian Kitainda for her assistance with protein purification and $O_2$ consumption measurements.

## Author contributions

All authors have given approval to the final version of the manuscript. S.S., T.V., Y.O., H.J.K., M.B., N.S., B.A.G., D.S.-D., K.M., and F.D. all contributed to aspects of the experiments, analyzed data, and wrote and edited the manuscript. S.S. purified, characterized, and prepared samples; S.S and T.V. processed data and built models; Y.O. performed genetic constructs; M.B. and D.S.-D. conducted computer modeling; H.J.K., N.S., and B.A.G. obtained MS data; and K.M. and F.D. managed the overall project and supervised the study.

## Competing interests

The authors declare no competing interests.
