## [Peer Review File · Nature Communications]

Reviewers' comments:

Reviewer #1 (Remarks to the Author):

This work reports for the first time a supercomplex involving a cbb3-type respiratory oxidase with its natural redox partners, bc1 plus cyt c2 and/or cyt cy. The work is novel and informative though it leaves many questions unanswered. The major drawback of the work is that it is not clear whether the supercomplex is simply an artificial construct resulting by fusing CcoP and PetC subunits together or whether this genetic manipulation just stabilizes an otherwise transient supercomplex. The authors should make a clear case for the physiological relevance of this supercomplex. This question extends to the placement of cyt cy in the supercomplex and the conclusion that cyt cy and cyt c2 interact differently with the hemes in CcoP.

Apart from the important question of the relevance of the supercomplex, the structures do provide an independent structure of a cbb3-type oxygen reductase (only one other structure is reported) as well as different conformational states of the bc1 complex. These data add to the significance of the work and justify its publication in this journal. The use of cross-linking to help in molecular modeling is also a useful addition.

Reviewer #2 (Remarks to the Author):

The authors combine mutagenesis to create fusion constructs of electron transport proteins with CryoEM and cross-linking mass spectrometry to produce structural models of respiratory supercomplex assemblies derived from Gram-negative bacteria.

Cross-linking studies of respiratory supercomplexes have previously been shown feasible, even in complex systems under conditions that support functional respiration. In addition, cross-linking mass spectrometry studies have been successful with live Gram-negative bacterial cells. These prior studies should be discussed in the present manuscript in the context of the studies performed here. Moreover, the authors propose inherent SC instability during isolation as a key limitation for structural characterization studies, leading to their efforts to employ fusion proteins to enable bipartite and tripartite supercomplex assembly formation. While that argument is likely correct, the authors do not speculate on what if any structural perturbations or artifacts arise from inclusion of fusion proteins that confine conformational space and stabilize the assemblies used for their CryoEM and XL MS studies. While the reported complementation of mutants lacking CIII and CIV is reassuring, this observation alone does not obviate structural perturbations or artefacts that may result from mutagenesis.

Moreover, a convincing piece of data that appears to be missing in this manuscript includes identification of analogous cross-linked peptides directly from cross-linked bacteria. Identification of these same cross-linked sites in respiring bacteria in the absence of the fusion protein linkers would greatly support the claim on page 17 that the: "...established salient structural features of two distinct respiratory electron transport pathways (membrane-confined and membrane-external) that operate between CIII2 and CIV in Gram-negative bacteria for the first time." Does cross-linking on cells result in increased intensity of SC bands in non-denaturing gels from WT cells? Discussion either way would be helpful. If so, this would appear to be a missed opportunity and its absence leaves the reader with uncertainty regarding the impact of fusion linkers on the derived SC structural models. Of course, a negative observation (i.e., lack of identification of the same linked sites from bacteria) would be inconclusive, but the successful observation of these links in wild type cells would be convincing.

FDR for cross-linked peptide searches is discussed and indicated to be <1% but the database used for cross-linked peptide searches is not explicitly defined and needs to be. Use of proteins listed in Table S2 as a database for XL-MS search would seem unwise since only selected bands were analyzed. Were aliquots of the purified complex samples digested and analyzed with mass spectrometry to establish what proteins were present in the sample prior to cross-linking? If so, these aspects need to be

defined and better discussed.

The rationale and utility of two different cross-linkers was unclear. Increased discussion would be helpful to guide readers. Also, the derived structural models and compatibility with links from either DMTMM or DSBU needs increased discussion. As DMTMM is a so-called zero-length linker, its use would normally be focused on obtaining tight distance constraints. Thus, the seemingly large distance constraint of 30 Å for DMTMM indicated on page 12 is puzzling and makes one wonder why a zero-length linker would be used here.

Some sort of summary supplemental figure illustrating all the observed links in Table S4 on proposed structures with a histogram of distances would be helpful in interpreting the results.

Reviewer #3 (Remarks to the Author):

This paper examines the cryo-EM structures of bc1-cbb3 type CIII2CIV respiratory supercomplex from Gram-negative bacteria, and expounds the structural characteristics of the electron transport pathways. These results are beneficial supplements to the current research field of respiratory chain, giving a referable example of combining cryo-EM and XL-MS in mechanism study. It might be useful if the author could provide the cross-validation results in validation part to eliminate the possibility of over-fitting. Genetically modified strain carrying a translational fusion between CIII2 and CIV enables the purification of SCs, but also increased the risk of artificial results. Detailed information might be required to compare the difference between modified-type and wild-type bacteria in respiratory activity and other possibly related characteristics. It is necessary to dispel the misgivings of artificial pseudomorph.

Response to the Reviewers

Reviewer #1

This work reports for the first time a supercomplex involving a cbb3-type respiratory oxidase with its natural redox partners, bc₁ plus cyt c₂ and/or cyt cy. The work is novel and informative though it leaves many questions unanswered.

- Thank you.

1- The major drawback of the work is that it is not clear whether the supercomplex is simply an artificial construct resulting by fusing CcoP and PetC subunits together or whether this genetic manipulation just stabilizes an otherwise transient supercomplex. The authors should make a clear case for the physiological relevance of this supercomplex. This question extends to the placement of cyt cy in the supercomplex and the conclusion that cyt cy and cyt c₂ interact differently with the hemes in CcoP.

- We agree that the studied super-complex (SC) is not “native” as it is engineered for increased stability to allow its study. We think that the genetic construct not only stabilized by increasing residency time of the interactions between bc₁ and cbb₃, but also allowed successful co-isolation of endogenous SC assembly partners, such as CcoH and cyt c_y. These components are barely detectable when bc₁ and cbb₃ are purified separately. The engineered SC is fully active (*in vivo* and *in vitro*), is produced in cells without any harmful effect, and supports growth under physiological conditions, as what might be expected with a “native” entity.

13 amino acid residues, spanning the fusion joint between the C-end of PetC and N-end of CcoP (no extra amino acid inserted in between), were not observed in our structure. When stretched, these 13 residues could span ~45Å, and the distance between the two residues at the end of PetC and beginning of CcoP is only ~21Å in our structure, suggesting that the fusion does not constrain the relative orientation between bc₁ and cbb₃ in the engineered SC. Other similar constructs with longer linkers (8-12AAs inserts, see the text) did not show any severe growth phenotypes. Altogether, these data further suggest that linking PetC to CcoP did not impair the native-like SC activity drastically.

We have now clearly stated in the text that the native-like SC structures are not to be taken as structures representing a plausible ‘native’ SC, which is not available.

2- Apart from the important question of the relevance of the supercomplex, the structures do provide an independent structure of a cbb3-type oxygen reductase (only one other structure is reported) as well as different conformational states of the bc₁ complex. These data add to the significance of the work and justify its publication in this journal. The use of cross-linking to help in molecular modeling is also a useful addition.

- Thank you.

**** For all three Reviewers (#1, #2 and #3) we wish to elaborate the findings that lead us to accept that the engineered SC is a “native-like” entity, and not an artificial artefact:**

1- The SC is produced and tolerated in active form *in vivo* to support anoxygenic photosynthetic (Ps) growth of a double mutant lacking both bc_1 and cbb_3 (**Fig. S1C**). In this species, anaerobic Ps growth requires an active bc_1 , and the onset of Ps growth needs an active cbb_3 of high O_2 affinity. Moreover, the engineered SC must interact productively with its electron carriers (cyt c_2 and c_3) to sustain physiological growth

2- A double mutant lacking both bc_1 and cbb_3 and producing SC exhibits Cox activity *in situ* under respiratory growth conditions, as demonstrated by the positive staining of colonies (**Fig. S1C**). This Cox activity is specific to cbb_3 as no other Cox exists in these cells, unlike others species.

3- Trace amounts of a ‘native’ entity of size, activity, and electrophoretic mobility comparable to those of the engineered SC is detectable in wild type membranes. We now provide additional data showing that this entity becomes more visible, though still unstable, using digitonin instead of DDM to disperse the membranes (**Additional data Fig. 1**; BN-PAGE/immunodetection).

4- The engineered SC is enzymatically active *in vitro* (in membranes and in purified state) for partial (*i.e.*, bc_1 or cbb_3) and coupled (*i.e.*, $bc_1 + cbb_3$) activities. These activities are similar to those of native bc_1 , cbb_3 , and a putative “native” SC of this type (**Fig. S1, I-K**).

5- The assembly component CcoH, which is not a part of the fusion construct, is found at the interface of isolated SCs. Similarly, upon addition *in vitro*, the native electron carrier cyt c_3 associates tightly with the interface of the SC via its membrane-anchor TMH, whereas a variant lacking this TMH does not (**Fig. S4 A,B**).

6- Mixing *in vitro* purified bc_1 and cbb_3 yields an entity of MW larger than the bc_1 or cbb_3 . This entity, although unstable, is distinguishable from the bc_1 or cbb_3 using glycerol gradients sedimentation (**Additional data Fig. 2**; Fractions analyzed by SDS/PAGE).

7- Mixing *in vitro* purified bc_1 and cbb_3 yields single particles of sizes and classes comparable to those seen with the engineered native-like SC, as depicted by negative staining EM (**Additional data Fig. 3**; EM micrographs).

- Based on these findings, we think that the engineered native-like SCs are likely to correspond to the stabilized versions of native SCs, and are not artificial artefacts. We were lucky to successfully obtain a stable SC that is functional under physiological conditions, allowing us to determine its structure and its interactions with the electron donors. This provides the proof of principle that a native-like functional SC is maintained in a Gram negative species, implying that such cells may naturally harbor a similar entity.

Reviewer #2:

The authors combine mutagenesis to create fusion constructs of electron transport proteins with CryoEM and cross-linking mass spectrometry to produce structural models of respiratory supercomplex assemblies derived from Gram-negative bacteria.

1- Cross-linking studies of respiratory supercomplexes have previously been shown feasible, even in complex systems under conditions that support functional respiration. In addition, cross-linking mass spectrometry studies have been successful with live Gram-negative bacterial cells. These prior studies should be discussed in the present manuscript in the context of the studies performed here.

- These studies are now discussed, and related references are added to the text.

2- Moreover, the authors propose inherent SC instability during isolation as a key limitation for structural characterization studies, leading to their efforts to employ fusion proteins to enable bipartite and tripartite supercomplex assembly formation. While that argument is likely correct, the authors do not speculate on what if any structural perturbations or artifacts arise from inclusion of fusion proteins that confine conformational space and stabilize the assemblies used for their CryoEM and XL MS studies. While the reported complementation of mutants lacking CIII and CIV is reassuring, this observation alone does not obviate structural perturbations or artefacts that may result from mutagenesis.

- Possible structural perturbations and artifacts due to the fusions are discussed.

The text is further clarified that the native-like SC structures reported here are not to be taken as plausible ‘native’ SCs. Please see also the *response to all Reviewers* above.

3- Moreover, a convincing piece of data that appears to be missing in this manuscript includes identification of analogous cross-linked peptides directly from cross-linked bacteria. Identification of these same cross-linked sites in respiring bacteria in the absence of the fusion protein linkers would greatly support the claim on page 17 that the: ...”established salient structural features of two distinct respiratory electron transport pathways (membrane-confined and membrane-external) that operate between CIII2 and CIV in Gram-negative bacteria for the first time.”

- This issue is further discussed in the text.

Identification of the same cross-linked sites in whole cells and in purified samples, if possible, will add additional support for authentication of this and other SCs (*see additional data Fig.1*; ~ 650 kDa SC) as native-like entities. However, the portions of SCs accessible to crosslinking appears to be limited *in vitro*, and they may be even less accessible in the crowded periplasm of whole cells. Since only a small number of crosslinks were obtained with purified samples, we fear that an even lower number might be obtained with whole cells and may not be conclusive, as also pointed out by the Reviewer.

4- Does cross-linking on cells result in increased intensity of SC bands in non-denaturing gels from WT cells? Discussion either way would be helpful. If so, this would appear to be a missed opportunity and its absence leaves the reader with uncertainty regarding the impact of fusion linkers on the derived SC structural models. Of course, a negative observation (i.e., lack of identification of the same linked sites from bacteria) would be inconclusive, but the successful observation of these links in wild type cells would be convincing.

- The issue is further discussed in the text.

The bands shown on **Fig. S1** correspond to in-gel activities (IGA); we do not know whether the SCs will remain active after *in vivo* crosslinking, even though a “not-crosslinked” subpopulation, or an alternate pathway may still sustain respiration. We suspect that the active SCs would still be unstable upon DDM extraction, and that *in vivo* crosslinking might affect the migration patterns of many entities, and not increase specifically the intensity of the 450 kDa band exclusively.

However, additional data pertinent to this point are provided. The occurrence in native membranes of an unstable entity of comparable MW to that of the fusion SC is more readily detectable on BN-PAGE using digitonin (instead of DDM) solubilized membranes (**Additional data Fig. 1**).

4- FDR for cross-linked peptide searches is discussed and indicated to be <1% but the database used for cross-linked peptide searches is not explicitly defined and needs to be. Use of proteins listed in Table S2 as a database for XL-MS search would seem unwise since only selected bands were analyzed. Were aliquots of the purified complex samples digested and analyzed with mass spectrometry to establish what proteins were present in the sample prior to cross-linking? If so, these aspects need to be defined and better discussed.

- The database used for XL-peptide searches is clearly defined in the Methods section. The database used for cross-linked peptides searches included all identified subunits (**Table S2**) cyt *c*₁, CcoP, CcoN, cyt *b*, CcoO, FeS protein, cyt *c*₂, the assembly components CcoH, CcoS and CcoQ as well as the 116 proteins from the cRAP database (www.thegpm.org; contaminant proteins commonly occurring in MS analyses) to increase the size of the decoy database to obtain reliable false discovery rates.

- Proteins present in the sample prior to cross linking were also determined using the full proteomic database for *Rhodobacter capsulatus* (see Methods). We used purified (affinity chromatography followed by size exclusion) samples for crosslinking experiments, and have not detected any additional protein beyond the usual abundant proteins seen as impurities.

5- The rationale and utility of two different cross-linkers was unclear. Increased discussion would be helpful to guide readers. Also, the derived structural models and compatibility with links from either DMTMM or DSBU needs increased discussion.

- The rationale and utility of two cross-linkers is better explained and discussed in the text. Two cross-linkers of different lengths and chemical properties were used to better accommodate the flexible parts of the SCs. These are they the cyt *c* domain of cyt *c*₂, the FeS-ED of the FeS protein, and the *bc*₁-*ccb*₃ interface as seen in different SC structures (SC-1A and SC-2A, see the text). We thought that an advantage of using DMTMM was not only its short length, but also additional coverage of areas that are not covered by other cross-links.

6- As DMTMM is a so-called zero-length linker, its use would normally be focused on obtaining tight distance constraints. Thus, the seemingly large distance constraint of 30 Å for DMTMM indicated on page 12 is puzzling and makes one wonder why a zero-length linker would be used here.

- Yes, DMTMM is considered to be a zero-length linker, excluding the Ca-Ca distances of the lengths of the side-chains. It seems that about 70% of the cross-links are within 17Å, and the remainder extend to distances up to 30 Å and beyond (see Leitner et al. PNAS, 2014, Fig. 3 with DMTMM in orange). We chose the value of 30 Å as the upper limit of distance to be used for docking in order to maximize the information content of the obtained cross-links. This upper limit

seems acceptable, as we restricted to 25 Å the “center to center” distances that separate the donor-acceptor pairs for productive electron transfer rate among the docking models.

7- Some sort of summary supplemental figure illustrating all the observed links in Table S4 on proposed structures with a histogram of distances would be helpful in interpreting the results.

- The previous **Fig. S4** is modified to include a panel summarizing the crosslinks and a related histogram. Please note that as multiple parts of the SC structure are mobile and flexible, only those pertinent to a specific conformation (SC-2A, see the text) are depicted.

Reviewer #3

1- This paper examines the cryo-EM structures of *bc1-cbb3* type CIII2CIV respiratory supercomplex from Gram-negative bacteria, and expounds the structural characteristics of the electron transport pathways. These results are beneficial supplements to the current research field of respiratory chain, giving a referable example of combining cryo-EM and XL-MS in mechanism study.

- Thank you.

2- It might be useful if the author could provide the cross-validation results in validation part to eliminate the possibility of over-fitting.

- The model-map cross-validation is now included into **Table S3** for all maps: row [Resolution (Model-map) at FSC 0.5 (Å)].

3- Genetically modified strain carrying a translational fusion between CIII2 and CIV enables the purification of SCs, but also increased the risk of artificial results.

- We now clearly stated in the text that the structural and functional properties of native-like fusion SCs are not to be taken as representing those of putative “native” SCs, which are unknown.

In respect to this point, please also see *Our response to all Reviewers* (above) and the additional experimental data provided.

4- Detailed information might be required to compare the difference between modified-type and wild-type bacteria in respiratory activity and other possibly related characteristics. It is necessary to dispel the misgivings of artificial pseudomorph.

- The species used here has multiple and overlapping respiratory pathways, with each branch being fully capable of supporting respiration. One branch contains *bc₁* and *cbb₃*, whereas the alternate branch (*i.e.*, Qox, quinol oxidase pathway) is independent of them. In order to dissect the respiratory activities *in vivo* one would need a strain lacking the alternate Qox branch, and the absence of *bc₁* and *cbb₃*. Unfortunately, such a triple mutant cannot grow by photosynthesis or respiration.

In wild type membranes, the individual complexes and super-complexes are thought to be in equilibrium with unknown and possibly variable ratios. We do not think that direct comparison of respiratory activities *in vivo* of modified vs wild type strains is readily and reliably feasible.

- In respect to ‘artificial pseudomorph’ please see *Our response to all Reviewers* above. We do not claim that the structure of the engineered native-like SC is representative of that of a putative “native” SC, which is unknown.

Additional data for the Reviewers

Figure 1. BN- PAGE-IGA and cyt b immunoblots. Comparison of CIV in-gel activities of wild type membranes solubilized with digitonin *versus* DDM.

Figure 2. Glycerol gradient sedimentation of a mixture of purified bc_1+cbb_3 (bottom panels) compared to a similar run with purified individual cbb_3 only (top panels).

Figure 3. Negative staining EM pictures of single particles and 2D classes of purified fusion SC (top panel) and a mixture of purified bc_1 and cbb_3 (bottom panel).

A) CIV (cbb₃) in-gel-activity of detergent-solubilized wild type membranes separated by 4-13% NativePAGE. Left: BN-PAGE, wild type (MT1131) membranes solubilized at different detergent:protein ratios. Protein bands migrate at higher MW with digitonin as compared to DDM. At 1:1 digitonin, the activity is smeared from ~350-600 kDa, indicating the presence of several species within this range. With increasing detergent:protein ratio, only the bands at the lower end of the smear are seen, suggesting that higher MW species collapse to lower MW species. **Right:** CN-PAGE, wild type and a bipartite fusion SC (*) are separated. As the gel run is not straight, separation between the two lanes is marked by a dashed line on the top part; the wild type activity band smeared towards the right. The location of the fusion SC bc₁-cbb₃ (see Fig. S1B of the text) is indicated by an arrow on aligned gels for comparison.

B) α-cyt b probed immunoblot of digitonin solubilized wild type (MT1131) membranes separated by Invitrogen 3-12% NativePAGE. Under appropriate conditions, a band running at MW between ~440-669 kDa (*) suggesting the presence of a bc₁-cbb₃ SC is observed upon over exposure of the film. An additional fainter band of even higher MW(?) of unknown identity is also seen. Note the very large amounts of individual bc₁ ~ 230kDa, as compared to the SCs (right), individual cbb₃ (not detected) also runs in this MW range.

Additional data Figure 1

Glycerol gradient sedimentation followed by SDS-PAGE of the mixed purified bc₁ and cbb₃ proteins (**bottom panel**, bc₁+ cbb₃) compared with cbb₃ alone run under the same conditions (**top panel**, cbb₃). 200 µg of proteins were loaded onto a 10-40% glycerol gradient in 50 mM Na-Phosphate, pH 8, 100 mM NaCl, 0.01% DDM, and sedimented by ultracentrifugation. 100 µl fractions were collected from top to bottom of the gradient, and 15 µl of selected fractions were analyzed by 12.5% SDS-PAGE and silver stained. Top panel shows that cbb₃ alone is predominantly found in fractions 16-18 and barely visible in fractions > 20. In the bc₁ + cbb₃ mixed sample, fractions 10-12 contain mostly bc₁, while 14-15 contain mostly cbb₃, and the fractions 18-24 contain both bc₁ and cbb₃, suggesting the occurrence of an SC of higher MW.

Additional data, Figure 2

bc₁-cbb₃ fused construct

native bc₁ + cbb₃

2D class averages of the bc₁-cbb₃ bipartite fused construct (1652 particles) versus the 1:1 mixed native proteins bc₁ and cbb₃ (1572 particles). The complete sets of class averages are shown, selected classes of the mixed sample resembling the size and shape of the fused complexes are highlighted with red boxes.

Additional data Figure 3

REVIEWERS' COMMENTS

Reviewer #1 (Remarks to the Author):

The authors have responded to my major criticisms very reasonably. There is a good chance that the SCs they stabilize by fusions are representative of what is physiologically relevant. The work presented is interesting and well presented and makes an excellent contribution to an area of growing interest. I favor publication in this journal.

Reviewer #3 (Remarks to the Author):

The major concern of our last review is that in what extent the bipartite and tripartite supercomplex they got after genetic engineering can represent the native state of respiratory chain in gram negative bacteria. We agreed that this attempt to stabilize the structure by specific site mutation is appealing, and for sure can answer many questions about the electron transfer pathway and the interactions among protein subunits, but the native role of this supercomplex is still very important.

This time, the authors made a great effort to explain the engineered SC is a "native-like" entity, rather than an artificial artefact. We notice that the engineered SC can rescue the double mutant lacking both bc1 and cbb3, and trace amount of the putative "native" entity is detected in both native gels and negative stain images, which are very similar to the engineered SC. The authors also claim very clearly that the engineered SC are not to be taken as the plausible "native" SC, which could hamper the value of this work, but still add valuable information to the understanding of respiratory chain.

Besides, we found a minor mistake in line 346 of the article. It seems the "tripartite" and "bipartite" should be interchanged.

Finally, we agree to publish this article in Nature Communication, for this article provides very valuable information about the electron transfer pathway of respiratory chain in gram negative bacteria.

Reviewer #4 (Remarks to the Author):

The authors use Cryo-EM and cross-linking with mass spectrometry to produce structural models of genetically engineered, functional respiratory complexes from Gram-negative bacteria. On the whole this is nice work providing structural information on bc1-cbb3 type respiratory supercomplexes. There are a few outstanding questions related to the cross-linking work detailed below.

Overall the discussion of the cross-linking results in the section, Interactions of cyt c2 and cyt cy with CIII2 CIV beginning on page 12, could be discussed more clearly so that readers have a better understanding of the rationale the authors had behind the cross-linking strategy they chose and the resulting data and its interpretation by the authors.

1 – Two different chemical cross-linkers were used although the rationale for this is not as clearly explained as it should be. The authors should clarify why they chose to use the carboxylic acid reactive cross-linker DMTMM as well as the primary amine reactive DSBU. DSBU is mentioned as having a longer spacer arm length, but the different reactive chemistries and the implications of this relative to the results should also be discussed.

2 – The authors utilize three different database search algorithms (MeroX, FindXL and MassAI) to analyze their LC-MS data and identify cross-linked peptide pairs. The rationale for this should also be explained in particular why the different algorithms were required, if they were, and why MeroX and FindXL were used for DMTMM while MeroX and MassAI were used for DSBU. They also mention that the results were filtered to an FDR of <1% but it is not clear if this is on the combined

results from all algorithms or individual results from each search? It is also not clear if the FDR threshold refers to cross-linked peptide spectral assignments (which can contain redundancy), or non-redundant cross-linked peptide sequences, or non-redundant residue pairs? It is important to specify the specifics of the FDR estimation in particular related to cross-linked peptide pair assignment as highlighted by Fisher and Rappsilber (<https://www.ncbi.nlm.nih.gov/pmc/articles/PMC5423704/>). Furthermore, scores and confidence levels resultant from the search algorithms should be included in the table of cross-linked peptides table S4. The authors should also include the unfiltered search results from all algorithms as supplementary tables or datasets so that readers can assess the complete results and how they were filtered.

3 – The authors provide a histogram of distances (presumably between alpha-carbons?) for a set of cross-links presented in FigS4C, while useful to get a sense of the overall fit of the XL-MS data with the structural models, it is difficult to see how these distances correspond to any specific links. It would be helpful to include distances along with the table of identified cross-links so that each distance can be directly related to a specific cross-link. Another option would be to display the cross-link bars on the figures where they are shown using a color gradient scale that changes with distance so over-length cross-links could be more clearly distinguished highlighting those regions thought to be disordered or flexible.

4 – In the methods section on Docking of cyt cy to CIII2CIV SC the authors mention using maximum distance thresholds of 30 angstroms for DMTMM and 35 angstroms for DSBU. Presumably there are Euclidean Ca-Ca distances? It should be specified. Also were these same thresholds applied for all other docking analyses? This also should be specified. How many resulting models from each of the docking analyses were considered for the cluster analysis? Top 10, 20, 100? Also the specific cross-links that could not satisfy the resulting models should be indicated somewhere instead of just providing a percentage of the data (i.e. 89% of the data on line 897).

Response to the Reviewers

Reviewer #1

The authors have responded to my major criticisms very reasonably. There is a good chance that the SCs they stabilize by fusions are representative of what is physiologically relevant. The work presented is interesting and well-presented and makes an excellent contribution to an area of growing interest. I favor publication in this journal.

- Thank you.

Reviewer #3:

1- The major concern of our last review is that in what extent the bipartite and tripartite supercomplex they got after genetic engineering can represent the native state of respiratory chain in gram negative bacteria. We agreed that this attempt to stabilize the structure by specific site mutation is appealing, and for sure can answer many questions about the electron transfer pathway and the interactions among protein subunits, but the native role of this supercomplex is still very important.

This time, the authors made a great effort to explain the engineered SC is a “native-like” entity, rather than an artificial artefact. We notice that the engineered SC can rescue the double mutant lacking both bc1 and cbb3, and trace amount of the putative “native” entity is detected in both native gels and negative stain images, which are very similar to the engineered SC. The authors also claim very clearly that the engineered SC are not to be taken as the plausible “native” SC, which could hamper the value of this work, but still add valuable information to the understanding of respiratory chain.

- Indeed, we concur. Thank you.

2- Besides, we found a minor mistake in line 346 of the article. It seems the “tripartite” and “bipartite” should be interchanged.

- We modified the corresponding sentence to further clarify it. Thank you.

3- Finally, we agree to publish this article in Nature Communication, for this article provides very valuable information about the electron transfer pathway of respiratory chain in gram negative bacteria.

- Thank you.

Reviewer #4:

The authors use Cryo-EM and cross-linking with mass spectrometry to produce structural models of genetically engineered, functional respiratory complexes from Gram-negative bacteria. On the whole this is nice work providing structural information on bc1-cbb3 type respiratory supercomplexes.

- Thank you.

There are a few outstanding questions related to the cross-linking work detailed below.

a- Overall the discussion of the cross-linking results in the section, Interactions of cyt c2 and cyt cy with CIII2 CIV beginning on page 12, could be discussed more clearly so that readers have a better understanding of the rationale the authors had behind the cross-linking strategy they chose and the resulting data and its interpretation by the authors.

- The section on p. 12, related to “Interactions of cyt c₂...” was rewritten to provide a better description of the rationale behind the cross-linking guided docking strategy chosen and the resulting data and its interpretation.

1 – Two different chemical cross-linkers were used although the rationale for this is not as clearly explained as it should be. The authors should clarify why they chose to use the carboxylic acid reactive cross-linker DMTMM as well as the primary amine reactive DSBU. DSBU is mentioned as having a longer spacer arm length, but the different reactive chemistries and the implications of this relative to the results should also be discussed.

- The rationale and utility of two cross-linkers are better explained and discussed in the text. Two points guided our choice:

1- various parts of the SC and its electron carriers have a limited number of membrane-external accessible lysines to be crosslinked with DSBU, and

2- the structure showed that parts of the SC and its electron carriers (cyt c_y, FeS-ED domain and CIV-CIII₂ interface) are flexible (see the structures SC-1A and SC-2A). Thus, the use of two different cross-linkers with different chemical properties and spacer lengths provided better coverage and tighter linker distances to improve docking by Patchdock.

2 – The authors utilize three different search algorithms (MeroX, FindXL and MassAI) to analyze their LC-MS data and identify cross-linked peptide pairs. The rationale for this should also be explained in particular why the different algorithms were required, if they were, and why MeroX and FindXL were used for DMTMM while MeroX and MassAI were used for DSBU.

- The main search engine used was MeroX, and yielded cross-linking results for both DMTMM and DSBU. The MeroX results were cross-validated to improve the level of confidence by using two other search engines for the same datasets. The FindXL search algorithm, which is limited to non-cleavable crosslinkers, was used for the non-cleavable DMTMM, while MassAI was used for the MS-cleavable DSBU data. These points are included in the Methods section, and the crosslinks of high level of confidence are listed in **Table S4**.

3- They also mention that the results were filtered to an FDR of <1% but it is not clear if this is on the combined results from all algorithms or individual results from each search? It is also not clear if the FDR threshold refers to cross-linked peptide spectral assignments (which can contain redundancy), or non-redundant cross-linked peptide sequences, or non-redundant residue pairs? It is important to specify the specifics of the FDR estimation in particular related to cross-linked peptide pair assignment as highlighted by Fisher and Rappsilber (<https://www.ncbi.nlm.nih.gov/pmc/articles/PMC5423704/>).

- The FDR filter does not refer to combined results. The MeroX results were filtered to a FDR of <1%, using the MeroX built-in cross-link spectral matches (XSM) false discovery rate (*i.e.*, based on cross-linked peptide spectral assignments). FDR calculations were not performed for MassAI or FindXL searches; results from these search engines were filtered by score, minimum number of fragments, intensity and other criteria as detailed in Methods. These searches provided cross-validations of the MeroX results for DSBU and DMTMM analyses, respectively. Only the cross-links of high confidence identified with two search engines in each case were used as distance restraints for XL-MS guided docking via Patchdock.

3a- Furthermore, scores and confidence levels resultant from the search algorithms should be included in the table of cross-linked peptides table S4. The authors should also include the unfiltered search results from all algorithms as supplementary tables or datasets so that readers can assess the complete results and how they were filtered.

The revised Table S4 includes scores from the main search engine MeroX, with the FDR threshold at < 1% as described in Methods. For the sake of clarity, the unfiltered search results are not included; all raw data are available in the PRIDE repository, and the search engines used are of public domain. If requested, the authors will gladly provide these data.

3 – The authors provide a histogram of distances (presumably between alpha-carbons?) for a set of cross-links presented in FigS4C, while useful to get a sense of the overall fit of the XL-MS data with the structural models, it is difficult to see how these distances correspond to any specific links. It would be helpful to include distances along with the table of identified cross-links so that each distance can be directly related to a specific cross-link. Another option would be to display the cross-link bars on the figures where they are shown using a color gradient scale that changes with distance so over-length cross-links could be more clearly distinguished highlighting those regions thought to be disordered or flexible.

The distances between the non-flexible parts of the SC structures are included in the revised **Table S4**, as requested.

4 – In the methods section on Docking of cyt cy to CIII2CIV SC the authors mention using maximum distance thresholds of 30 angstroms for DMTMM and 35 angstroms for DSBU. Presumably there are Euclidean Ca-Ca distances? It should be specified. Also were these same thresholds applied for all other docking analyses? This also should be specified.

Yes, these are *Euclidean Ca-Ca distances* and yes the same thresholds were applied to all docking analyses. These points are now specified in the Methods section.

4a- How many resulting models from each of the docking analyses were considered for the cluster analysis? Top 10, 20, 100?

For each docking analysis, the top 50 resulting models were considered, and this is specified in the Methods section. For clarity a smaller number of these models are visualized in appropriate figures as indicated in their legends.

4b- Also the specific cross-links that could not satisfy the resulting models should be indicated somewhere instead of just providing a percentage of the data (i.e. 89% of the data on line 897).

The cross-links that could not be satisfied in each model resulting from Patchdock are variable; it is not always the same XLs that are not satisfied in all models. It would be cumbersome to list the unsatisfied XL for each model, but a % value reflects the degree of confidence on the cluster of models thus defined (e.g., for 89%, of the 9 XLs used, only 1 was unsatisfied in any given model, yielding 8/9 satisfaction).